# Drop impact printing

Chandantaru Dey Modak 1, Arvind Kumar[1,2], Abinash Tripathy [1,3] & Prosenjit Sen [1✉]

Hydrodynamic collapse of a central air-cavity during the recoil phase of droplet impact on a superhydrophobic sieve leads to satellite-free generation of a single droplet through the sieve. Two modes of cavity formation and droplet ejection have been observed and explained. The volume of the generated droplet scales with the pore size. Based on this phenomenon, we propose a drop-on-demand printing technique. Despite significant advancements in inkjet technology, enhancement in mass-loading and particle-size have been limited due to clogging of the printhead nozzle. By replacing the nozzle with a sieve, we demonstrate printing of nanoparticle suspension with 71% mass-loading. Comparatively large particles of 20 μm diameter are dispensed in droplets of ~80 μm diameter. Printing is performed for surface tension as low as 32 mNm$^{-1}$ and viscosity as high as 33 mPa•s. In comparison to existing techniques, this way of printing is widely accessible as it is significantly simple and economical.

[1] Centre for Nano Science and Engineering, Indian Institute of Science, Bangalore, Karnataka 560012, India. [2] Present address: Australian Institute of Bioengineering and Nanotechnology, The University of Queensland, Brisbane, QLD 4072, Australia. [3] Present address: Department of Mechanical and Process Engineering, ETH Zurich, Sonneggstrasse 3, CH-8092 Zurich, Switzerland. ✉email: prosenjits@iisc.ac.in

Dispensing small droplets is of great research interest because of its numerous applications in electronic industry, medical science, automobiles, and rapid prototyping. However, printing small droplets is challenging due to the dominance of surface tension at length scales smaller than the capillary length $\sqrt{\gamma \rho^{-1} g^{-1}}$ (where $\gamma$ is surface tension, $\rho$ is density, and $g$ is gravitational acceleration). For generating small droplets, surface tension force is usually overcome by applying an external force (e.g., electrical, thermal, or acoustic). Inkjet printers are well established for conventional printing. However, certain applications require printing of liquids containing biological samples, biopolymers, and micro-/nanoparticles. Since conventional inkjet technology is not designed to work with such inks, modification of the printhead nozzle is required to maintain the desired resolution, accuracy, and widespread applicability[1–5]. This in turn increases the cost and complexity of the setup. Emergence of other printing techniques[6–9] has also taken place using acoustic[10], electro-hydrodynamic (EHD)[11–13], laser-assisted[14], or microfluidics[15]-based designs. Use of complex technology in these techniques prohibitively increases their setup and operational cost[16,17]. Hence, the availability of these printing techniques for research, development, and other scientific purposes has not yet been prevalent in limited resource scenarios.

Most microdroplet printing technologies use a nozzle-based dispensing configuration[17] with integrated actuators and a complex drive/control system[18–20]. The nozzle primarily focuses the applied force and hence determines the ejected droplet size. In these technologies, two main disadvantages of satellite droplets[21–23] and nozzle clogging[9,24,25] mostly remain unaddressed. Conventional inkjet printer nozzles with a lot of system modifications have used material jetting. However, these modifications indirectly increase complexity and cost[26,27]. Satellite drops are unwanted products of the droplet-formation process that reduces pattern quality. Nozzle clogging predominantly happens due to solvent evaporation while attempting to print inks with either higher mass loading or large particles. Nozzle clogging is mostly destructive requiring replacement of the expensive nozzle. These issues severely restrict nozzle-based printing to use liquids with either a limited range of properties[22,23] or definite mass loading[9,24,25,28].

This report describes a new way of printing that mitigates these issues. Based on the impact of a droplet on a superhydrophobic sieve, the setup is exceptionally simple. Cavity collapse during the recoil phase leads to satellite-free generation of a single microdroplet. This printing technique has certain advantages with respect to conventional inkjet printing technique in terms of (i) capability to print droplets with high mass loading, (ii) capability to print large particles (i.e., comparable to the dispensed droplet diameter), (iii) large range of printed droplet volumes (diameter varying from ~42 μm to ~960 μm), (iv) satellite-free printing, and (v) simple setup and low operational cost. Printing resolution and accuracies are not as good as some other techniques (e.g., EHD printing). However, the high resolution and accuracies for drop-impact printing have been demonstrated for the above-mentioned printing conditions[12,29,30], which are not easily attainable by other techniques. This report focuses on demonstrating a generic printer, and hence, the capability to print liquids with very high viscosities is not as good as acoustophoretic printing[6]. Drop-impact printing can achieve very high throughput in terms of dispensed volumes. This is due to the technique's ability to print droplets with a significantly wider range of diameters (42–960 μm). Further, the individual sieve supports multiple simultaneous droplet impacts, providing parallelization.

By replacing the nozzle with a sieve, we demonstrate printing with high mass loading (71%) and large particle size (20 μm). Apart from this, the technique is cost-effective, compact in size, easy to operate, and allows instant reconfiguration for different microdroplet sizes. Using this setup, the paper reports printing of various inks for different applications. In addition to traditional applications, this technique can be used for (1) ceramics-based 3D printing[31] for dental prostheses[32] and architectural modeling[33], (2) dispensing biological samples for single-cell applications, 3D organ printing[34], and (3) printing for electronic applications[5,9]. Apart from its versatility, this technique is remarkably affordable and hence will make drop-on-demand printing widely accessible.

## Results

**Cavity-collapse-driven single-microdroplet ejection.** The outcome of drop impact on a superhydrophobic sieve is determined by the balance between the dynamic pressure ($\sim \rho U_0^2$) of the impinging droplet and the breakthrough pressure ($\sim 4\gamma L^{-1}$) of the sieve[35,36]. Here, $U_0$ is the impact velocity, $\rho$ is the density, $\gamma$ is surface tension, and $L$ is the size of the pore as shown in Fig. 1a. In our impact experiments, water droplets of diameter, $D_o \sim 2.56$ mm were released from different heights varying from 2 cm to 5 cm. Droplet ejection was captured using a high-speed camera (Photron FastCam) operating at frame rates as high as 75,000 frames per second. Using a superhydrophobic sieve #0.009 (refer to Supplementary Table 1 for geometrical parameters), the possibility of single-droplet generation was evaluated. At lower impact velocities ($U_0 = 69$ cm s$^{-1}$, Weber number ($We$) = 17), the liquid failed to penetrate the mesh (Supplementary Fig. 1a) as the breakthrough pressure was higher than the dynamic pressure.

As the impact velocity increased, a regime of single-droplet ejection was observed (Supplementary Movie 1 and Supplementary Fig. 2). However, the microdroplet creation was not observed during the impact phase. The impact pressure ($\sim \rho U_0^2$) was not enough for ejection of liquid through the pore and its subsequent separation by Rayleigh–Plateau instability[19,37]. Only during the recoil phase, microdroplet generation was observed. Hence, this phenomenon has been termed as recoil ejection[35]. Without identifying a physical cause, prior literature has attributed this ejection to increase in local pressures during the retraction phase[35]. On further increasing the impact velocity ($U_0 = 83$ cm s$^{-1}$, $We = 25$), microdroplet ejection was observed during the spreading phase. This however led to generation of multiple droplets (Supplementary Fig. 1b). Henceforth, we focus on the single-droplet generation by recoil ejection.

Experimentally, we observed the formation of an air cavity during interface retraction and its collapse just prior to the recoil ejection as seen in Fig. 1b. Formation of air cavity has been previously reported for its impact on a flat hydrophobic surface[38]. Droplet impact creates capillary waves, which leads to formation of a cylindrical air cavity trapped between the retracting interface. Motion of the interface causes the cavity to collapse, and the kinetic energy of the fluid converges along the axis of collapse. This inertial focusing causes the interface velocity to diverge[39]. Local dynamic pressure ($\sim \rho U^2$, $U$ is the velocity of the collapsing cavity walls/interface. When the cavity collapses, the local velocity of the collapsing front is higher than the impact velocity) at the collapsing front becomes much larger than the impact dynamic pressure ($\sim \rho U_0^2$). On flat surfaces, the resulting hydrodynamic singularity causes ejection of a narrow high-speed jet[38]. For impact on superhydrophobic sieves, the pore limits the lateral extent of the collapsing cavity. It also sets the lateral boundary for the interface motion resulting from the cavity collapse. The sieve topography changes the collapse dynamics, and a single microdroplet is ejected as seen in Supplementary Movie 2 and Supplementary Fig. 3a, b. This mode where the cavity is

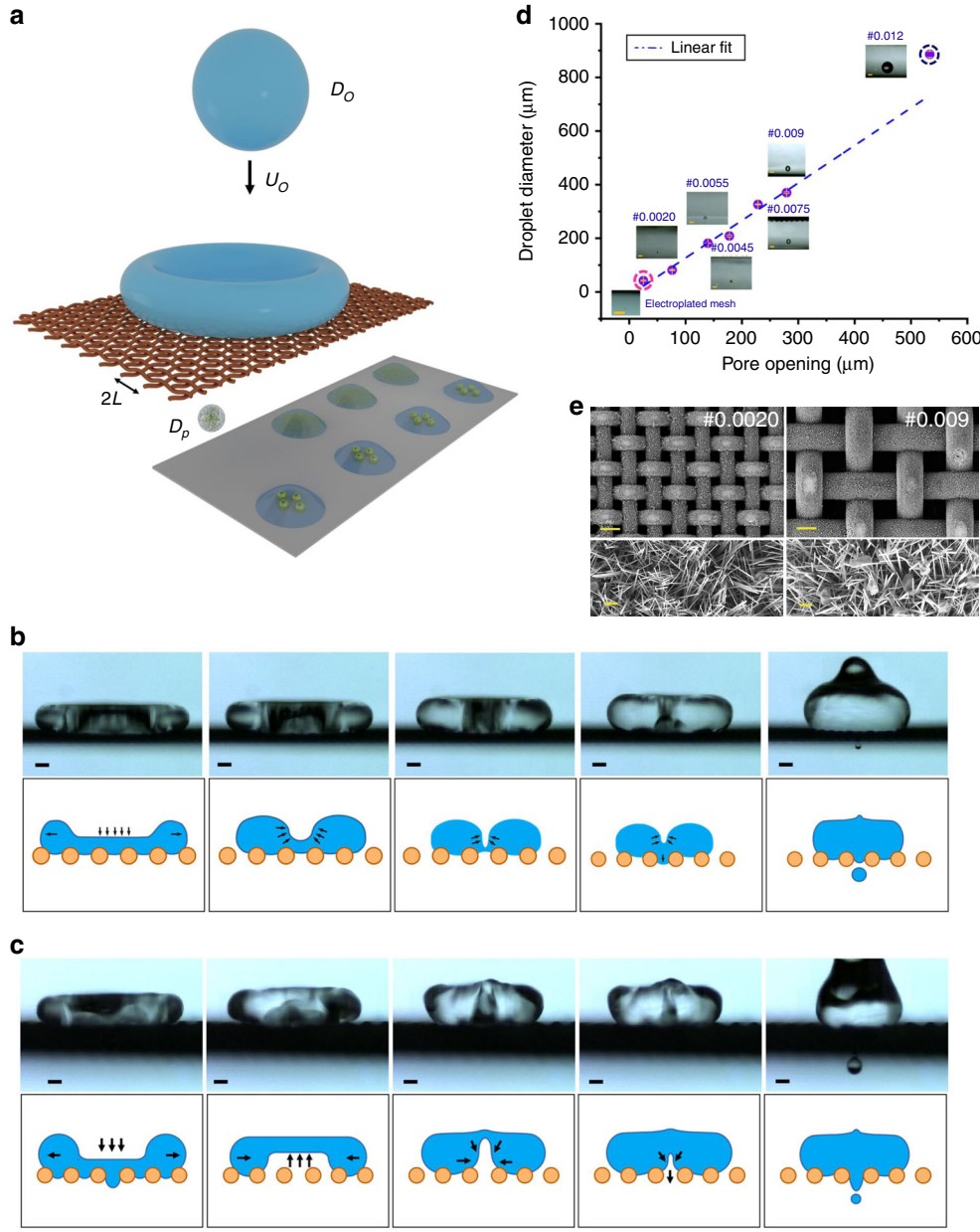

**Fig. 1 Mechanism and explanation of drop-impact printing technique. a** Schematic illustration showing the drop-impact setup, a droplet (diameter $D_o$, velocity $U_o$) impacting on a superhydrophobic sieve (pore opening, $L$) to eject out a single smaller droplet (diameter $D_p$). The impacting drop gives rise to two modes of single-droplet ejection. **b** Impact cavity (IC) and (**c**) recoil cavity (RC). Scale bar: 200 μm. The time-lapsed images and schematic illustration for IC and RC modes show the mechanism of cavity formation and collapse using sieve #0.0045 with 65% glycerol water droplet and sieve #0.009 with pure water droplet, respectively. The drop-impact printing technique was explored in terms of the smallest ejected droplet that can be generated. **d** Shows a plot between water droplet diameter versus pore opening, and the insets show the corresponding patterned droplet (scale bar: 100 μm). Superhydrophobic sieves with different pore openings were used starting from sieve type #0.012 (pore opening L: 533.4 μm, wire diameter W: 304.8 μm) to #0.0020 (pore opening L: 76.2 μm, wire diameter W: 50.8 μm) and electroplated mesh (pore opening L: 25.2 μm, wire diameter W: 101.2 μm) marked in blue-dotted circle. **e** Scanning electron microscopy (SEM) images of sieves #0.009 and #0.0020 (scale bar: 100 μm, magnified image scale bar: 2 μm).

formed during the initial impact has been termed impact-cavity (IC) mode.

In our experiments with sieve #0.009, we observe a new mode of cavity formation as shown in Supplementary Movie 3 and Fig. 1c. The liquid penetrating the meshes during impact is observed to recoil back and move up[40]. This is due to the surface energy stored in the penetrating liquid ($\sim\gamma L^2$). The liquid moving up from the pores completely fills the initial cavity (impact cavity) formed during the spreading phase. Interestingly, the interface

recoiling from the pores does not stop at the top surface of the mesh. The interface is observed to move up through the droplet, and a new cavity is formed. This cavity has been termed as recoil cavity (RC). Collapse of cavities formed by both IC and RC mode leads to the single-droplet generation that we use for printing applications.

**Satellite-free droplet ejection.** Satellite drops are an artifact of breaking an ejected stream into droplets due to Rayleigh

instability. Hence, a common strategy for eliminating satellite droplets has been to attain separation with shorter neck lengths. Nozzle-based printers commonly use actuation waveforms with positive- and negative- pressure pulses. An initial positive pulse is used to eject the liquid, whereas the negative pulse pulls back the bulk liquid to enable quick separation[41]. Conceptually, these schemes attempt to create a short pulse of focused energy at the tip.

In recoil ejection, we naturally observe satellite-free droplet creation. Here, the collapse of the cavity focuses the kinetic energy. The pore limits the resultant interface motion and distributes this energy over a length scale of the pore ($\sim L$). Beyond this length scale, the dynamic pressure quickly falls to bulk values. The droplet separation is further aided by the bulk flow, which during the collapse (recoil phase), is pointed away from the surface as observed in simulation results (Supplementary Fig. 4and Supplementary Note 1). In conjunction, both these effects inherently create the conditions generated by the complex pulse train in nozzle-based printers. The importance of recoil ejection for satellite-free droplet generation became apparent when hydrophobic meshes were used. In hydrophobic meshes, droplets were generated in impact-ejection mode only. In this mode, where inertial focusing is absent, longer necks and satellite droplets were observed.

**Drop-on-demand printing**. Water droplets of different diameters were dispensed using sieves with different pore openings (see Fig. 1e for SEM images of the sieve). Supplementary Table 2 shows the range of a dimensionless number for which water droplet printing was carried out with varying pore openings. Figure 1d shows the plot of the ejected droplet diameter (measured using ImageJ[42]) as a function of the pore opening. The size of the ejected droplet was proportional to the pore opening, $D_p = 0.88^*L^{1.07}$ (except for sieve #0.012). For sieves other than #0.012, the liquid from initial penetration was able to retract back, and the whole microdroplet volume was from the liquid penetrating the mesh after the cavity collapse. We name it collapse-penetration mode (CPM). Different possible outcomes of droplet impact are shown in Supplementary Fig. 5.

Compared to other sieves, sieve #0.012 ejects out higher droplet volume for its pore opening. Sieve #0.012 has the largest pore opening. Unlike other meshes, the liquid from initial (impact) penetration is unable to retract back (Supplementary Fig. 6a–d). This liquid combines with the liquid brought in by the cavity collapse during recoil, and leads to a higher ejection volume (Supplementary Fig. 6e, f and Supplementary Movie 4). This mode of microdroplet creation has been named impact-penetration mode (IPM). Although the droplet volume for sieve type #0.012 is a bit higher as compared to other sieves, it still gives satellite-free dispensing. Thus, this technique provides the capability of printing a wide range of single droplets of diameter ranging from 94 μm to 926 μm (Supplementary Movie 5).

To eject out smaller drops (<94 μm), we need a mesh with smaller pore sizes. Mesh type #0.0020 (pore opening—76.2 μm, wire diameter—50.8 μm) is the mesh with the smallest pore opening that is available from the manufacturer. Copper was electroplated on mesh type #0.0020 to reduce the pore size. We were able to reduce the pore opening to ~32 μm. The copper electroplating process schematic and parameters are shown in Supplementary Fig. 7. The electroplated mesh was processed like other meshes to obtain superhydrophobic pores. SEM of the electroplated and etched mesh is shown in Supplementary Fig. 8a, b. The contact angles on the electroplated and etched super-hydrophobic mesh were found to be 92° ± 2° and 161° ± 4°,

respectively (Supplementary Fig. 8c, d). The drop-impact printing experiment was performed on the electroplated superhydrophobic mesh, and a drop size of ~42 μm in diameter was obtained (Fig. 1d and Supplementary Fig. 8e). Supplementary Movie 6 shows the printing of an ~42 μm droplet. This approach uses a simple electroplating process to reduce the pore opening of the mesh for printing droplets of smaller size, and further increases the resolution of drop-impact printing.

The capability to print a broad range of liquids was validated by using Newtonian (other than water), and non-Newtonian fluids of varying viscosities and surface tensions. Viscosity was varied by adding glycerol to water (Supplementary Note 2). Surface tension variation was obtained by adding polyethylene glycol (PEG) or ethanol to water. Supplementary Table 3 shows the properties of these liquids. As seen in Fig. 2a, b, single-droplet printing was possible for viscosity as high as 33 mPa.s and surface tension as low as 32 mNm⁻¹. The ejected droplet diameter was mostly independent of varying viscosity (Fig. 2a) and surface tension (Fig. 2b). However, transition from the CPM to IPM mode of ejection led to a slight increase in volume. For the largest pore opening (sieve #0.012), IPM mode of droplet creation was observed for all values of surface tension and viscosity. For other meshes, transition to IPM mode of ejection was observed at higher viscosities. Similarly, a transition from CPM to IPM mode of ejection was observed for lower surface tension values. Finally, droplet impact was used to print viscoelastic liquid (xanthan gum and water) of varying concentrations (1–10%, volume percent (v/v)). Single-droplet printing was observed up to viscosity 20 mPa.s (Supplementary Fig. 9). Thereafter, printing of high-viscosity ink is not trivial, and further improvements (augmentation) in the setup are required.

Figure 2c shows the printable region for Newtonian fluid in the terms of Ohnesorge number ($Oh = \mu(\sqrt{\rho\gamma L})^{-1}$, $\mu$: viscosity) and Reynolds number ($Re = \rho U_0 D_0 \mu^{-1}$). As compared to traditional drop-on-demand printers, the current technique can print using a wider range of fluid properties[43,44] (detailed review of ink palette used by existing printing techniques for different applications has been shown in Supplementary Table 4). Figure 2d compares the $Z$ number ($Oh^{-1}$) for our technique with traditional drop-on-demand printers. Drop-impact printer can print for $Z$ values varying from 3 to 200, which is significantly better than the reported range of 1–14 for commonly used techniques. Below $Z < 3$, viscous force is high, so the liquid is unable to penetrate the sieve, and a maximum of Z~200 corresponds to the water drop using sieve #0.012.

Which mode (IPM or CPM) is observed is determined by a competition between the different timescales pertaining to droplet impact and liquid penetration. The penetrated interface is able to recoil back if its dynamics is faster (the timescale is shorter) than that of the impacting droplet. Impact dynamics of the parent droplet is dominated by inertia with a timescale of $\tau_d \sim \sqrt{\rho D^3 \gamma^{-1}}$. The timescale of liquid penetration and retraction is determined by liquid inertia and viscosity. In a purely inertial regime, the timescale of the penetrated interface is given by $\tau_i \sim \sqrt{\rho L^3 \gamma^{-1}}$. CPM will be observable when the nondimensional ratio of these timescales (timescale factor TSF $\sim \sqrt{L^3 D^{-3}}$) is smaller than a critical value. In the viscous regime, the timescale of the penetrated interface is given by $\tau_v \sim \mu W \gamma^{-1}$ (where $\mu$ is viscosity and $W$ is the width of mesh wire). The crossover from the inertial to viscous regime happens when the timescale to set up viscous flows in the pore ($\sim \rho L^2 \mu^{-1}$) is smaller than the inertial timescale ($\sqrt{\rho L^3 \gamma^{-1}}$). This implies that viscous effects become dominant above a crossover Ohnesorge number $Oh_{cr}$. The equations can be rearranged to get a common timescale

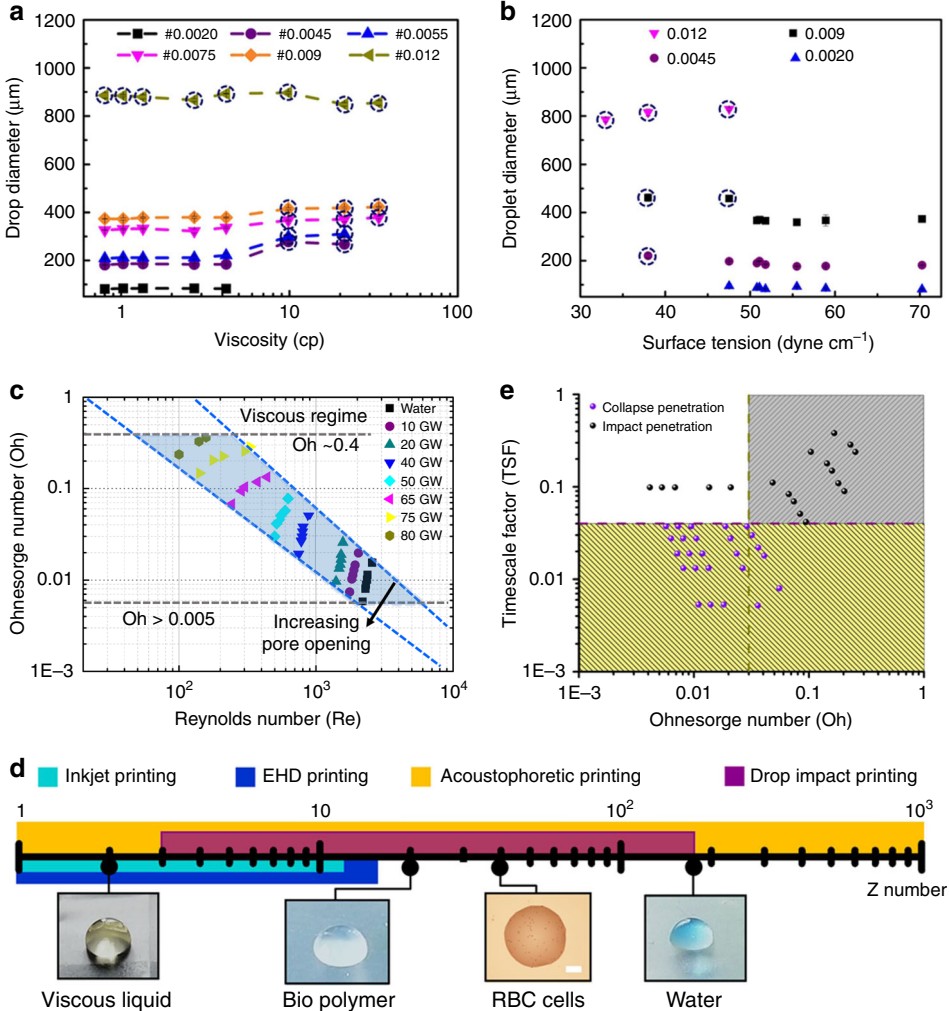

**Fig. 2 Parametric studies showing the capabilities of drop-impact printing technique.** The extent of viscous liquid and low surface tension liquid printing was explored using glycerol water solution, polyethylene glycol (PEG) water solution, and ethanol-water solution. The ejected droplet diameter was plotted with (**a**) liquid viscosity and (**b**) liquid surface tension for a sieve with different pore openings. **c** The printable regime was observed in the plot between Ohnesorge and Reynolds number. The light-blue shaded part shows the printable region of drop-impact printing technique. The range gives us an idea of the extent of different liquids that can be used for printing. **d** The broad range of liquids is shown in terms of Z number with inset images showing the different liquid drops that can be printed. The drop-impact printing technique (shown with purple color bar) was compared to inkjet printing, electrohydrodynamic (EHD) printing and acoustophoretic printing represented with turquoise, blue, and yellow bars, respectively (scale bar: 100 μm). **e** The mechanism of different ejection modes was explained based on a timescale factor with varying Ohnesorge number. The critical Ohnesorge number that ensures transition from the inertial to viscous regime was 0.03, and the time- scale factor value that defines the transition from collapse-penetration mode (CPM) to impact-penetration mode (IPM) was found to be 0.04.

factor given by

$$\text{TSF} = f\left(\frac{Oh \times (W \times L^{-1})}{Oh_{cr} \times (W \times L^{-1})_{cr}}\right) \times \sqrt{L^3 D^{-3}} \quad (1)$$

$$f(x) = 1, \text{for } Oh < Oh_{cr}$$

$$f(x) = x, \text{for } Oh \geq Oh_{cr}.$$

For calculation of TSF, it is necessary to identify the critical $Oh$ beyond which viscous forces are no more negligible. We identify $Oh_{cr}$ by considering the sieve with the largest pore (#0.009), which transitions from CPM to IPM ($Oh_{cr} \approx 0.03$). The enhanced role of viscosity is also evident from Weber number required for the ejection of a single droplet (Supplementary Fig. 10). TSF is plotted in Fig. 2e. For sieve #0.012, a large mesh size leads to IPM ejection even in the inertial regime. For our

experiments, a critical TSF of 0.04 seems to separate the two regimes well.

**Printing of large particles**. In conventional nozzle-based inkjet printers, the nozzle diameter limits the particle size that can be printed. It has been reported that for printing of suspensions, the printer nozzle diameter should be 100 times greater than the particle size, otherwise nozzle clogging may occur[45] (Supplementary Table 5). Printing of larger particles is required for cell suspensions, functionalized microbeads, and 3D micro-particle structuring for dental prosthetics. By eliminating the nozzle, the drop-impact printing performed considerably better. Even with sieve #0.0020 having the smallest pore opening of 76.2 μm, we could print 20-μm polystyrene beads without clogging (Supplementary Table 6 for nanoparticle sizes). Figure 3a illustrates the broad range of particle size that can be printed using drop-impact printing. The capability to handle

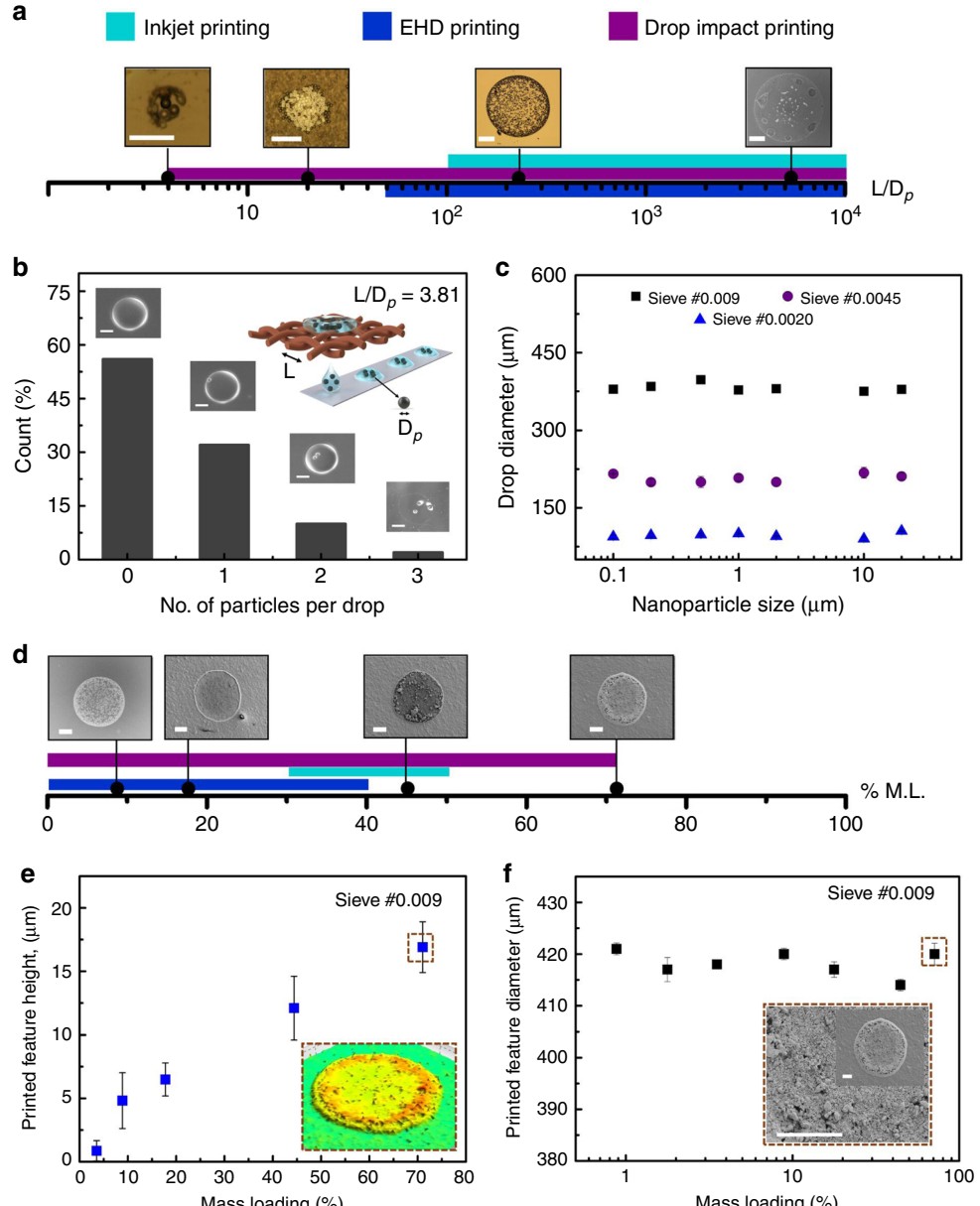

**Fig. 3 Clogging-free printing with a large particle size and higher mass loading printing.** The clogging-free printing was demonstrated based on the ability to print a large particle size and high mass loading suspensions. **a** The larger particle size printing ability was shown in a linear $L/D_p$ chart with the inset showing a different printed particle size for different $L/D_p$ (scale bar: 100 μm) ratios. $L/D_p$ can be as low as 3.81 for drop-impact printing, which is significantly smaller as compared to inkjet and electrohydrodynamic (EHD) printing. **b** The percent count to print a single and multiple beads in a drop is demonstrated. The probability of single-bead capturing in a single drop (80-μm diameter) was found to be 32%. The inset shows the number of beads in a single drop (scale bar: 100 μm). **c** Further, the printed droplet diameter with varying particle size is shown. The droplet diameter was independent of different particle-size suspensions. **d** The linear chart shows that as high as 71% mass loading suspension solution printing is possible using drop-impact printing as compared to inkjet and EHD printing. The inset shows the scanning electron microscopy (SEM) image of a printed droplet for different mass loading (scale bar: 100 μm). **e** The printed feature height is shown with varying mass loading (print substrate—glass). The inset shows the printed droplet with 71% mass loading having a base diameter of 990 μm (sieve used—#0.009). **f** Further, the printed droplet diameter was plotted with varying mass loading (inset image scale bar: 100 μm). The printed drop size was found to be independent with an increase in mass loading. Insets in both figures (**e**) and (**f**) show a higher mass loading printed drop.

different particle sizes is quantified as a ratio of nozzle to particle diameter. For drop-impact printing, this ratio goes down to 4 from the traditional known value of 100. The significant advancement can be attributed to the sieve configuration where the sample liquid is only in intermittent contact (~10 ms) with the nozzle (sieve pore). This eliminates the probability of nozzle clogging due to particle agglomeration. We further quantify the probability of single-bead trapping.

The probability of getting a single 20-μm polystyrene bead in a 0.268-nL volume drop is 32% (Fig. 3b).

The viscosity of dilute suspensions is known to vary linearly with concentration. However, the rheology of suspensions with higher concentration of nanoparticles is complicated due to complex particle–particle and particle–fluid interactions. Rheological behavior of such suspensions is expected to vary not only with mass loading but also the size of the suspended particles. We have studied

the effect of changes in suspended particle sizes on microdroplet ejection. As seen in Fig. 3c, droplet diameter does not vary with particle size in the suspensions for a given mass loading of 9%.

**Printing with high mass loading.** Nozzle clogging also depends on the mass loading. Printing inks with higher mass loading is beneficial as it reduces the number of reprints required for achieving higher thickness. As loading increases, the viscosity increases, which makes jetting of suspensions difficult. However, the major challenge of printing suspensions is due to the enhanced nozzle clogging from preferential drying of the solvent at the nozzle tip[24]. Previous reports state that the clogging can be reduced by using a proper dispersion agent. Even with these measures, printing could be achieved for mass loadings of only up to ~45%[46,47] (refer Supplementary Table 5 for detailed review).

We carried out experiments to estimate the maximum mass loading that can be achieved using sieve #0.009. The ink was formulated using different concentrations of $ZrO_2$ nanoparticle dispersed in 10 vol% PEG (Supplementary Note 2). Figure 3d shows SEM images of ejected droplet for different mass loadings. The illustration shows the drop-impact printing technique's range to print high mass loading as compared to other technologies. We were able to achieve repeatable microdroplet generation for a maximum mass loading of 71%. In these experiments, the mesh was slightly tilted to ensure that the impacting droplet did not settle on the mesh after impact. The droplets rolled down to the sealed ink reservoir will be recycled (Supplementary Fig. 18) and can be used again for printing. Although the entire ink will not be recoverable due to evaporation, according to previous studies, the rate of evaporation in a completely sealed setup is very low[48,49] (volume loss percent is 0.01% for a timescale of 100 ms) and its effect on ink concentration (% mass loading) will be significant after continuously printing for a very long duration. A completely sealed setup for eliminating evaporation will bring in complexity in terms of precise environment control and hence increase the cost. However, such a setup will not be a general requirement. Such a system will be necessary for expensive inks with high vapor pressure. We believe that even though the current system does not eliminate evaporation completely, it will be sufficient for most applications.

Using 71% mass loading, we were able to achieve deposition thickness of 16.9 μm in a single print (Fig. 3e) using mesh #0.009 (drop-base diameter—990 μm). As expected, with an increase in mass loading, we observed an increase in the deposition thickness. The ejected droplet diameter was found to be approximately the same with variation in mass loading (Fig. 3f). The average mass loading of the remaining solid mass for a batch of 50 printed drops after drying was found to be 66% ± 1.5%. This discrepancy of 5% in terms of mass loading may be due to the settling of nanoparticles inside the reservoir (syringe) or pipe during the printing process. At the highest mass loading, a small amount of residues was left on the sieve by the impacting droplets. In our experiments, this affected the lifetime of the impact location on the sieve to a limited number of impacts. Unlike a clogged nozzle, in our case, the residues can be easily removed by washing with a mild jet of deionized (DI) water followed by $N_2$ purging (Supplementary Fig. 11 and Supplementary Note 3), and the same mesh can be used for printing again.

**Printing accuracy.** The printing accuracy was evaluated in terms of droplet-size consistency and droplet exit angle. The droplet-size accuracy was measured by dispensing an array of 50 droplets of aqueous silver nanoparticles (4% v/v) through sieve #0.009 and #0.0020. The deposited droplets were heated at 90 °C for 4 h, and the size was measured from optical images using ImageJ software. As seen in Supplementary Fig. 12, the deposited droplets are monodispersed with sizes of 559 ± 11 μm and 83 ± 2 μm for sieves #0.009 and #0.0020, respectively.

The droplet exit angle was determined for sieves with different pore openings by using the images extracted from the high-speed videos. The droplet ejects out with an angle due to the absence of concentricity that imparts a horizontal velocity component to the ejected droplet. As the mesh is repositioned with respect to the syringe, we found different droplet-ejection angles. However, if the mesh is not moved, the impact process continues to eject microdroplets with the same angle. This helps us in obtaining sufficient accuracy for printing when the substrate is kept at a distance of 1.5 mm from the mesh (#0.012 and #0.0045). Positioning error was estimated by printing multiple drops. Supplementary Fig. 13 plots the deviation in the position while we attempted to print droplets along a straight line. The worst-case deviation was ∼ 30 μm in the lateral direction and ∼ 10 μm in the longitudinal direction (for meshes #0.012 and #0.0045). It is interesting to note that this is significantly smaller than the size of the droplets that can be printed with the respective meshes (Fig. 1d). This shows the ability of the technique to print a drop with comparative accuracy when compared to commercial printers. Even though our current setup does not incorporate prealignment of the impinging drop with the mesh pore, developing such a mechanism is possible using a camera and a motorized stage. To summarize the above discussion, a detailed review showing critical parameters like resolution, accuracy, ink properties, and cost, has been shown for drop-impact printing technique comparing it with the existing printing techniques in Supplementary Table 7. The comparison shows the critical points where drop-impact printing technique has an advantage over the existing printing techniques.

**Printing for biological applications.** In biological science, room-temperature printing of microarrays (bacteria, DNA, cells, and proteins) for gene-expression analysis, single-cell printing for basic biological cell studies, and biopolymer printings are of paramount interest. The present technique was tested for printing a smaller volume of biosamples and molecules. We performed single-drop printing of red blood cell (RBC) suspension. The RBC cells of varying concentrations were printed on a glass slide. Figure 4a shows the printed drop of different cell concentrations. The concentration of cell solutions varied from $4 \times 10^4$ to $62 \times 10^4$ cells per $mm^3$. In addition to this, we investigate the number of cells per droplet with varying cell concentrations (Fig. 4b). The sample data are for 50 drops for both sieve #0.009 and #0.0045. The analysis revealed that as concentrations increased, the number of cells per droplet increases. Also, the cells remain isolated within the droplets. This gives us the benefit of using a very small sample volume that will be isolated from each other, and also reduces the time required for pipetting and placing samples. Further, the study was extended to print a single cell (MDA-MB-231) in a single drop of volume 0.268 nL (Fig. 4c). The present technology of printing large-cell solutions exists, but the nozzle clogging still remains a major challenge[50,51]. Thus, this technique provides us an effective solution for clogging-free printing of large cells for different applications.

One of the other ways of doing cell culture-based studies is by patterning. The culture substrate is patterned with different wettability. The simplest way to do is to change the surface chemistry, making superhydrophobic and superhydrophilic arrays. The drop-impact printing can also be used for making such gradient surfaces. DMEM liquid was used as printing ink in our case, and arrays of DMEM drops were printed on Teflon-coated substrate (Fig. 4d.1). Upon drying, the DMEM drops became hydrophilic, and the rest of the Teflon-coated surface remained hydrophobic, thus making a wettability gradient. When the MDA-MB-231 cell solution was allowed to flow over the gradient surface,

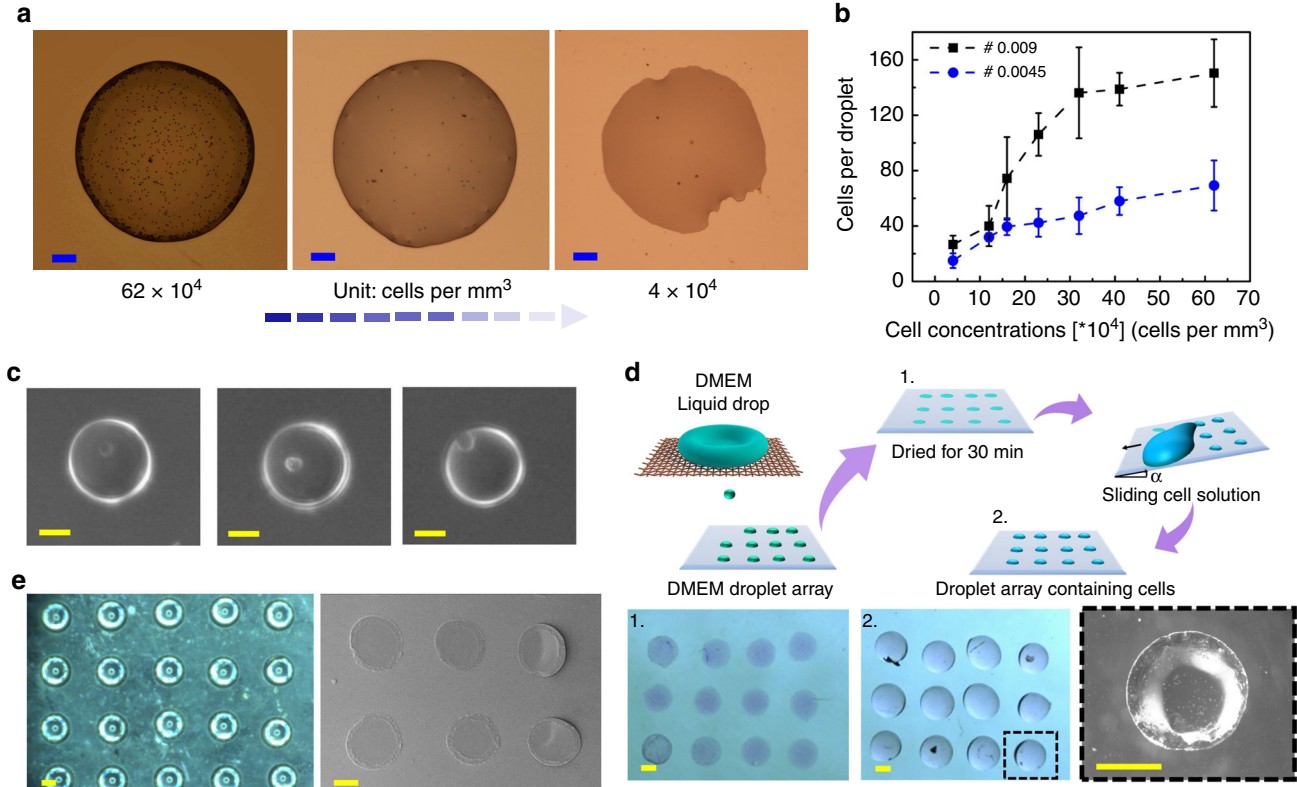

**Fig. 4 Drop-impact printing of biological solutions and biopolymers. a** Microscopic images of a single droplet patterned using cell (RBC)-laden PBS solution of different concentrations, scale bar: 50 μm. The cells are contained in an isolated droplet of volume 26 nL patterned using mesh type #0.009. **b** The number of cells per droplet for varying cell concentrations was examined for mesh types #0.009 and #0.0045. The single-cell printing was further demonstrated using drop-impact technique. **c** Single cells (MDA-MB-231) of average size ~17 μm were trapped in a 0.268-nL single drop. The drops were collected on an oil-coated glass slide. The concentration of cell solution was kept at $50 \times 10^4$ cells per mm³ (scale bar: 50 μm). **d** Illustration showing printed DMEM droplet arrays using drop- impact printing technique. (1) Shows printed DMEM droplets on a hydrophobic Teflon surface. (2) Shows the arrays of MDA-MB-231 cells containing droplets after cell solution swipe, and a magnified image of a printed droplet containing cells (scale bar: 500 μm). Beside this, the technique's ability was explored by using biopolymeric viscoelastic liquid (0.0125 g per mL polyacrylic acid mixed in water) for 3D printing applications. **e** The large patterned microposts of 875 μm diameter and 2 μm height were printed on APTES-coated glass slides and the corresponding scanning electron microscopy (SEM) image (scale bar: 400 μm).

the solution was trapped within the hydrophilic area (Fig. 4d.2, d.3). This technique is very useful for biological inks that are prone to contamination or have a shorter decay time[52]. Hence, this drop-impact printing technique provides us a new way to make such gradient surfaces without modulating the surface chemistry.

In addition, to realize the possibility of printing viscous bioink for 3D printing applications, polyacrylic acid was used as a model-printing liquid. Polyacrylic acid 1.25% (weight percent (w/w)) was used for printing droplet volume of 0.4 μL (948 μm diameter) in the form of a micropost. Figure 4e shows the optical and SEM images of a micropost created using polyacrylic acid polymer. Once the droplets are deposited on the (3-aminopropyl) triethoxysilane (APTES)-coated glass slide, it is kept at normal environment for curing. After curing, a polyacrylic micropost of diameter 875 μm and height of 2 μm was obtained. This result proves the versatility of the drop- impact printing technique to print a micron-size polymeric micropost. Not only it reduces the processing time, but it is also cost-effective and provides more flexibility.

**Printing for electronic applications**. Conducting lines were printed using aqueous solutions of silver-ink and poly(3,4-ethylenedioxythiophene) polystyrene sulfonate (PEDOT:PSS) polymer (Supplementary Note 2). Formation of a line requires deposition of subsequent droplets at an optimum displacement.

Too close a placement can lead to pattern widening, whereas too far a placement of subsequent drops will lead to discontinuity. The process is shown in Supplementary Fig. 14. The line was printed using sieve #0.009 and a droplet spacing of 150–200 μm. The droplet after it touches the substrate, first spreads and then oscillates. The combined effect of spreading and oscillation ensures the merging with the neighboring droplet after it lands. The concentration of silver ink was first optimized to get good conductivity with single- layer printing (Supplementary Fig. 15). At the optimized silver concentration of 4% (v/v), further printing demonstrations were shown.

Figure 5a shows the silver line of width 450 μm, length 2.5 mm, and average height of 0.655 μm. Figure 5b shows the PEDOT:PSS line with dimension 450 μm × 2.5 mm × 2.1 μm. The magnified image shows proper curing of polymer. The resistance of silver was found to be 31 Ω, and for PEDOT:PSS was 2.7 kΩ (Fig. 5c). This optimization plays an important role in printing-based applications and varies for different printing liquids. With this understanding, a diode was fabricated using silver and PEDOT: PSS line printed on a glass substrate. Figure 5d.1 and d.2 show the schematic diagram of the device and SEM image of the junction, respectively. The IV characteristics in Fig. 5d.3 show the diode characteristics of the fabricated device.

Finally, interesting demonstrations, including printed connections for LED on a flexible tape (Fig. 5e), large-area droplet array

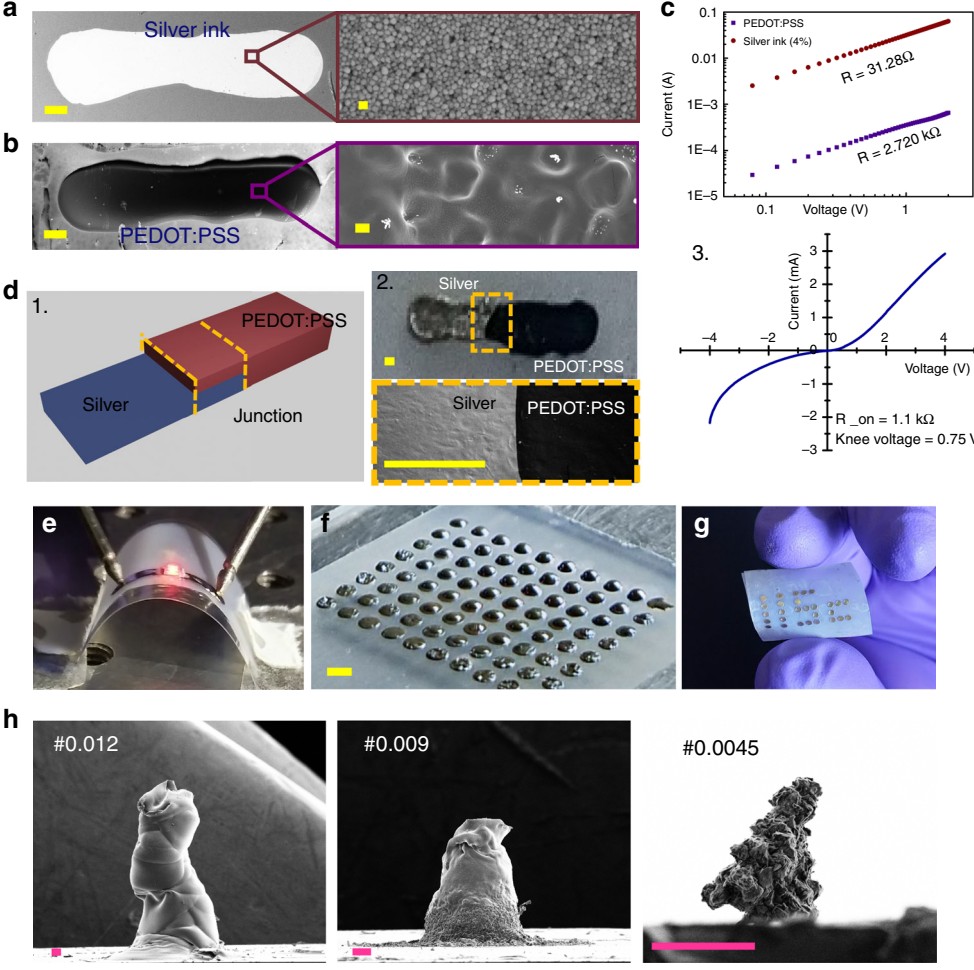

**Fig. 5 Printing of electrically conducting materials for large-area fabrication and flexible electronics applications.** Room-temperature printing of (**a**) silver ink (4% (v/v)) conductive line (scale bar: 200 μm) and the corresponding scanning electron microscopy (SEM) image (scale bar: 100 nm). **b** PEDOT:PSS-printed line (scale bar: 200 μm) and the corresponding SEM image showing the connectivity (scale bar: 20 μm). **c** IV characteristics of both silver and PEDOT:PSS-conducting lines. **d** (1) Silver ink and PEDOT:PSS were further used to form a junction to show the capability of the technique for electronic applications. (2) Optical microscopic and the SEM image showing the junction (scale bar: 250 μm). (3) In addition, IV characteristic was performed for the junction to check the connectivity. Further, as a demonstration (**e**) two silver-conducting lines are connected using drop- impact printing technique, and the voltage is applied at both ends to show the glowing LED. **f** Large-area droplet patterning (scale bar: 1 mm), **g** flexible printing, and **h** 3D pillars printed using sieves with different pore openings have been shown to demonstrate the wide applicability of this technique (scale bar: 100 μm).

(Fig. 5f and Supplementary Movie 7), printing of letters on a flexible substrate (Fig. 5g), and 3D-printed ZrO₂ pillars using sieves with different pore openings (Fig. 5h), are presented. Supplementary Movie 8 shows drop-by-drop printing of a micropillar using sieve #0.012. In addition, we demonstrate the possibility of scaling the printing process through multiple drop impacts on a single sieve #0.009 (Supplementary Movie 9). The main advantage of drop-impact printing is easy handling and cost-effective large-area printing (Fig. 5f and Supplementary Fig. 16).

## Discussion

In conclusion, this work presents a new drop-on-demand printing technique with a simple design and hence requires low setup cost. Use of a superhydrophobic sieve instead of a complex nozzle further reduces operational cost. Recoil ejection driven by the cavity-collapse singularity leads to satellite- free ejection of single droplets. The technique is found to generate monodisperse droplets. Further, this technique can handle a wide variety of printing solutions for different applications. As the contact between the sieve and the liquid is only for a limited duration of impact, this technique excels in printing of large particles and suspensions

with high mass loading. It does not require any electric, magnetic, or wave forces, except a pump that will pump the liquid. This work presents an easily accessible approach to generate picolitrer-to-microlitre-volume droplets for different applications like bio-culture, electronic printing, and functional material structuring.

## Methods

**Nanowire fabrication (superhydrophobic sieve).** Copper sieve of different pore openings and wire diameters was purchased from Copper TWPinc, USA. The growth of nanowires on copper surface was achieved by immersing copper for 15 min in an aqueous solution of 2.5 mol L⁻¹ sodium hydroxide and 0.1 mol L⁻¹ ammonium persulfate at room temperature[36]. The nanostructured surface was further dipped in 1H,1H,2H,2H-perfluorooctyltriethoxysilane solution overnight to achieve superhydrophobicity having water contact angle ~159° and CAH <5° (Supplementary Fig. 17).

**Experimental setup.** The printing setup along with the recycle unit is shown in Supplementary Fig. 18. Also, a video of the operational printing setup is shown in Supplementary movie 10. The superhydrophobic copper sieve of different pore openings (76.2–533.4 μm) having an area of 6 cm² was clamped from both the ends. The high-speed imaging was performed (Photron FastCam) from one side keeping the diffused LED light source opposite to it. The impacting droplet was generated from a 1-mL syringe using a syringe pump, generating droplets of size

2.55 ± 0.5 mm. Teflon or APTES-coated glass slides are used to collect the ejected droplet underneath the mesh at a distance of 1 mm.

## Data availability

All the data used in this paper and supplementary information are available upon reasonable request from the corresponding author.

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

## Acknowledgements

The authors would like to acknowledge Mr. Kritank Kalyan, project assistant, CeNSE, IISc Bangalore for MATLAB Code and Solidworks assistance, and Ms. Saba Tasneem, project assistant, CeNSE, IISc Bangalore, for helping in electroplating process. The authors thank Ms. Aditi Jain for providing the MDA-MB-231 cells used in the experiments, and Prof. M. M. Nayak for providing the peristaltic pump. A.T. would like to thank Prof. Dr. Thomas Schutzius, D-MAVT, ETH, Zurich, for his valuable inputs. All the authors would like to thank the Department of Science and Technology and Ministry of Electronics and Information Technology, Government of India for financial support.

## Author contributions

A.K., A.T., and P.S. designed the research; A.T. and P.S. provided scientific guidance; C.D.M., A.K., and A.T. performed experiments; C.D.M. and A.T. analyzed the data; C.D.M., A.K., A.T., and P.S. set up the model and wrote the paper. All authors have read and approved the final version of the paper.

## Competing interests

The authors declare no competing interests.
