## [Peer Review File · Nature Communications]

Reviewers' comments:

Reviewer #1 (Remarks to the Author):

The paper presents an interesting study on the development of a new technology for printing, using drop ejection after impact of millimetric water drop on a sieve to produce microdrops in a reproducible manner. The paper is solid and complete, since it provides a fundamental understanding of the phenomenon (drop ejection after impact on a sieve/mesh), as well as demonstrate the use for a variety of applications, such as printing of cell suspensions or conducting lines for electronic applications.

The reference to state of the art of current printing technology is well highlighted in Figure 3, where the potential of the technology is addressed, by showing the range of printable drops in terms of the Z number ($=1/Oh$).

In addition to that, the paper reads well, and I really enjoyed it, which I consider a valuable plus.

In conclusion, the paper is well prepared, and I have no doubt recommending publication of the paper, which is bringing forward a new idea in the field of liquid-solid interaction, paving the way for a new technology.

I have only one minor request.

- The authors should discuss the issue of sieve contamination: what happens if particles remain on the sieve as a result of multiple impacts? How many times can the sieve be re-used, before e.g. cleaning is required? This issue should be discussed in the text.

Reviewer #2 (Remarks to the Author):

In their manuscript titled "Drop Impact Printing", Modak et al. demonstrate a new printing method that utilizes the energy of droplet impact to generate small droplets through a hydrophobic sieve. As the authors correctly state, this process itself is not new (reference 31), but has not been analyzed for single droplet ejection and on-demand droplet patterning. The main and enabling novelties, as claimed by the authors, are the simplicity of the process, the achievable range of properties, and a high mass loading, among others.

While I do think that the presented process has potential, my major concern is that the authors' claims on the capabilities of existing processes and the capabilities of their process in comparison are not justified. In summary:

- In the introduction, the authors mention several, highly interesting, competing processes. Throughout the manuscript, however, they limit the comparison of their process to material jetting ("inkjet printing"). In some aspects, the presented process outperforms material jetting, but not in all aspects, some of which are critical. This includes, for example, the resolution, scalability, and actual "3D" printing capability (rather than the demonstrated 2D patterning).

- The authors claim to have introduced a new "3D printing" technique. While I generally believe that the process could be suitable for 3D applications, there are no such demonstrations and I have concerns that the process, in its current state, is not suitable for 3D printing. This is due to the positioning accuracy that is low compared to the droplet size (i.e., the average deviation from the targeted position is higher than the droplet size). I suggest removing all such claims or demonstrate actual 3D structures (with the same feature size and resolution as claimed).

- The presented process produces a lot of excess material, as only a fraction of the initially (large) droplet is eventually deposited onto the surface. I could not find the exact number in the manuscript, but would estimate it to be as high as 1/1'000 of the volume. Even if this material can be recycled after the printing process (which only works for few inks), it renders the process highly inefficient, if not unsuitable, for numerous inks and applications, many of which the authors claim to be "printable", such as cell-laden inks. It is simply infeasible (both monetarily and from a synthesis perspective) to produce 1'000x the amount of cell-laden ink that is needed for the final part.

- The materials palette is not as broad as claimed by the authors, when compared to processes other than material jetting. In fact, it is much smaller than that of, e.g., acoustophoretic printing.

- One of the claimed, major advantages is the printing of inks with high mass loadings. However, some of

the mass is (stochastically?) sieved out by the mesh, which a) clogs the sieve (and thereby diminishes one of the acclaimed advantages over other processes), b) reduces the effective mass loading, and c) reduces the repeatability.

I, thereby, recommend major revisions with an emphasis on proving the claims in question quantitatively or with specific references; or removing these claims (which, however, reduces the novelty and impact of the work).

In detail:

- The authors state that their process “outperforms conventional inkjet technique in most aspects.”. Besides the investigated factors, could the authors comment on other, important, metrics, such as the resolution (I believe inkjet printing can go below 10 μm , whereas the current process seems to be limited to 70 μm), throughput volume per time per nozzle (in comparison to inkjet printing), the scalability (inkjet printing can be scaled to hundreds of nozzles with minimal distance between the nozzles), and the positional accuracy?
- Lots of questionable statements are made without proof. E.g., “However, newer applications require printing of more exotic inks”: What are newer applications (as compared to which “old” applications)? And what are more exotic inks (as compared to “less” exotic inks)? This is just one example, I recommend carefully reviewing all such statements throughout the manuscript.
- The authors state that inkjet printing (more accurately and commonly referred to as material jetting) “fails to provide the resolution, accuracy and widespread applicability”. To my understanding, material jetting provides higher resolutions and higher accuracies as the process proposed.
- In turn, the authors claim that alternative printing techniques that use, e.g., acoustic forces or microfluidics, to overcome these aforementioned “limitations” are “restricted to a selected community”. I strongly disagree with this statement and believe that some of these techniques have incredible (and even higher) potential. For example, acoustophoretic printing can eject (“mass loaded”) droplets over significantly larger viscosity ranges than any other process, including the process proposed.
- What do the authors mean by nozzles with “complicated actuators”?
- The authors claim that other techniques are limited because of clogging of “expensive nozzles”. To my understanding, “cheap” nozzles are commonly and commercially available down to (inner) diameter sizes of 10 μm or smaller (which is smaller than what the current process can produce). I do not agree with this argument.
- The authors state a whole list of issue with all existing technologies and claim that their technology overcomes “these issues”, which leaves the impression that their technique is better than each and every of the existing techniques. I believe that their technique has some advantages, but also disadvantages over existing technologies and recommend reviewing all such statements.
- The same goes for statements like “[...] outperforms conventional inkjet technique in most aspects.” and others. Such statements are highly vague. Please also provide specific proof on all such statements.
- The authors claim that their process is unique in terms of the mass loading it can achieve (71%). Can the authors state what the (theoretical and practical) maximum values of (all of) the other processes are, they compare their process to?
- How do the authors generate the initial droplet? Please provide a detailed description.
- The authors say that their process is “cost-effective, compact in size”, when compared to existing methods. Can the authors quantify these statements, e.g., what is the size and cost of the complete set-up, and compare it to the other, competing methods they refer to? I believe that this is a very bold statement, which needs to be put into relation quantitatively. Can the authors add the droplet generation hardware to the schematic (and possibly add a photo)? Is the droplet generation hardware part of this “cost-effective, compact in size”-evaluation (which it should be)?
- The sieve is weaved in different, vertical planes (Figure 1e). This creates a strong directionality of the holes or “channels” away from being orthogonal to the plane of the sieve. How does this effect the direction of the generated, small droplet? Are the authors taking this into account when they eject the “big” droplet onto the surface, i.e., do the authors know which hole will be dispensing the “small” droplet?
- When looking at the paper that initially discovered the penetration of drops through superhydrophobic meshes (reference 31) and the methods in the presented work, it seems like there is a very narrow range in which (single) droplets can be reliably ejected (“impact cavity mode”). First, I assume it takes significant effort to find this regime for every new material, which stands in contrast to the simplicity of the approach

the authors claim? Second, does this not limit the materials palette of significantly (which stands in contrast with the versatility the authors claim)?

- The materials the authors claim to be ejectable differ in more properties than only the “viscosities and surface tensions” the authors investigate. A full rheological analysis is necessary.
- Can the authors show examples of the ejection of these specific materials (i.e., ceramic and cell-laden inks)? Can the authors show that, in particular, the cells in the cell-laden inks survive the ejection process?
- What happens to the excess material? What is the ratio of the volume of the initial droplet to the ejected droplet?
- The presented drop impact printing covers materials with Z numbers ranging from 2 to 200 (Figure 2d). This does not outperform inkjet printing on the lower Z number regime. Further, acoustophoretic printing, for example, covers materials with Z numbers ranging from 10⁻³ to 10³, which is significantly higher. Could the authors generate a more complete comparison in Figure 2e that shows not just inkjet and drop impact printing, but also all the other methods the authors claim are outperformed with their approach?
- In the manuscript, the authors say that “Drop-impact printer can print for Z values varying from 3 to 200”, but Figure 2d shows a range of 2 to 200. Please correct.
- Figure 3a: Shouldn't the green bar go down to 10² rather than the 10³, where it is currently (based on the text “100 times greater than the particle size”)? How do these values compare to the other methods mentioned in the introduction of the manuscript? Could the authors add bars for these methods (or, alternatively, remove the claim that they outperform these methods)?
- The authors say that they can eject particles with diameters of up to ¼ of the sieve pores. The authors further claim that they can eject inks with mass loadings of up to 71% (Figure 3). Can the authors quantify the particle loading (or mass) ratio of the ink before and after extrusion?
- Further, can the authors quantify the particle loading (or mass) ratio of a significant number of ejected drops to show that it is consistent (and that the sieve doesn't filter the particles or prevent them from ejection)?
- Clogging is often a function of time. Can the authors quantitatively show that they can maintain ejection of the high mass loading inks through the same pore over an extended amount of time (to maintain comparability with existing, nozzle-based processes)?
- The authors state “Unlike clogged nozzle, for our case the residues can be easily removed by washing with a secondary liquid.”. Do the authors mean that this would happen during the printing process? Also, I assume that this strongly depends on the ejected materials. Many materials might cure onto the sieve and form a strong bond that cannot easily be washed away.
- An droplet displacement error of 90 µm at a print height of 1 mm is significant, when the droplet diameter is 82 µm, and contradicts the “high accuracy” claim. Can the authors compare this number to other, existing processes (including inkjet printing and other techniques)?
- Please review the grammar and style throughout the manuscript.

Reviewer #3 (Remarks to the Author):

In this manuscript, the authors have proposed a new method for printing using droplet impact and a hydrophobic sieve. The authors showed that this approach could address the drawbacks of inject printing. I should assess this article in two aspect: one on novelty of this method and its promise and one on the fundamental aspect of this work.

The authors rationalizes that formation of satellite droplets and clogging are two drawback of inject printing approach and this new method could address these drawback. I would encourage the authors to think more carefully before making this claim. In this approach, the generated droplet has volume of 100 times less than the impact droplet. What will happen to the rest of the droplet? It will be all repelled and should be somehow recycled. This is much worse that any inject printing approach in which the droplet sizes and flow rate could be accurately controlled. The second drawback of the inject printing is clogging. The clogging problem will be much worse in the sieve compared to a short nozzle in inject printing. By appropriate surface coating of the nozzle, this problem could be addressed. Thus, this new proposed method does not offer any advantage over inject printing and in some aspects is worse.

On the fundamental aspect of this proposed approach, the authors have completely neglected the hammer

pressure developed at stagnation point. This pressure could be few orders of magnitude higher than dynamic pressure. There is no new fundamental understanding in this work. I would suggest the authors to correct the fundamental argument in this work and just focus on their observation rather than claiming its superiority over inject printing.

Response to reviewers' comments

Manuscript ID: NCOMMS-19-33513

“Drop Impact Printing”

We thank the editor and the reviewers for carefully reading our manuscript and providing insightful comments. We believe these comments will improve the overall quality of the manuscript. Here, we provide a point-to-point response to these comments. In doing so and in implementing the changes in the revised manuscript, we have paid particular attention to the reviewers' comments. For clarity, the reviewers' comments are in **blue**, and the changes we have made are highlighted in **yellow** in the revised manuscript.

Reviewer #1 (Remarks to the Author):

The paper presents an interesting study on the development of a new technology for printing, using drop ejection after impact of millimetric water drop on a sieve to produce microdrops in a reproducible manner.

The paper is solid and complete, since it provides a fundamental understanding of the phenomenon (drop ejection after impact on a sieve/mesh), as well as demonstrate the use for a variety of applications, such as printing of cell suspensions or conducting lines for electronic applications.

The reference to state of the art of current printing technology is well highlighted in Figure 3, where the potential of the technology is addressed, by showing the range of printable drops in terms of the Z number ($=1/Oh$).

In addition to that, the paper reads well, and I really enjoyed it, which I consider a valuable plus.

In conclusion, the paper is well prepared, and I have no doubt recommending publication of the paper, which is bringing forward a new idea in the field of liquid-solid interaction, paving the way for a new technology.

I have only one minor request.

- The authors should discuss the issue of sieve contamination: what happens if particles remain on the sieve as a result of multiple impacts? How many times can the sieve be re-used, before e.g. cleaning is required? This issue should be discussed in the text.

Response:

We would like to thank the reviewer for appreciating our work. We agree with the reviewer that there will be some degree of contamination during the drop impact printing. In order to examine that, we performed drop impact experiments using ZrO₂ nanoparticles (44.4%, w/w) dispersed in a mixture of Ethylene Glycol and water in the ratio 1:10. Drop impact using this solution was carried out for approximately 1000 times at one place using the mesh #0.012 coated with Teflon instead of stearic acid (Please refer Table S1 for the geometrical parameters of the mesh). After the experiments, we observed some degree of contamination on the spot where drop impacted the mesh as seen in Figure R1a. The contamination was easily removed with a mild jet of deionized (DI) water. After washing, the samples were purged in N₂ (Figure R1b,c). In order to verify that the superhydrophobicity of the mesh was retained after the wash, we quantified its wettability through measurement of contact angle and contact angle hysteresis (Figure R1d,e). Measured contact angle was $157^{\circ} \pm 3^{\circ}$ and hysteresis was $< 5^{\circ}$ (contact angle hysteresis was calculated from the advancing and receding angle of the droplet at the onset when the droplet starts to slide) satisfying the conditions of superhydrophobicity. Hence, authors believe that although there will be some sieve contamination, the same can be resolved with a mild water wash.

Figure R1: Sieve contamination characterization. (a) pinned nanoparticles on superhydrophobic sieve (#0.012) after approximately 1000 drop impacts. (b) cleaning process of sieve after contamination (water jet impact and N₂ purging). (c) Mesh after cleaning. (d) Static contact angle of water droplet on cleaned sieve and (e) Water droplet repellence test by impacting drops on superhydrophobic cleaned mesh.

Reviewer #2 (Remarks to the Author):

In their manuscript titled “Drop Impact Printing”, Modak et al. demonstrate a new printing method that utilizes the energy of droplet impact to generate small droplets through a hydrophobic sieve. As the authors correctly state, this process itself is not new (reference 31) but has not been analysed for single droplet ejection and on-demand droplet patterning. The main and enabling novelties, as claimed by the authors, are the simplicity of the process, the achievable range of properties, and a high mass loading, among others.

While I do think that the presented process has potential, my major concern is that the authors’ claims on the capabilities of existing processes and the capabilities of their process in comparison are not justified.

In summary:

- In the introduction, the authors mention several, highly interesting, competing processes. Throughout the manuscript, however, they limit the comparison of their process to material jetting (“inkjet printing”). In some aspects, the presented process outperforms material jetting, but not in all aspects, some of which are critical. This includes, for example, the resolution, scalability, and actual “3D” printing capability (rather than the demonstrated 2D patterning).

Response: We would like to thank the reviewer for the valuable inputs. The reviewer has asked about the capability of the present technique in terms of **resolution**, **scalability** and **3D printing**. The three aspects of the technique are discussed one by one in the paragraphs below.

Resolution: Mesh type #0.0020 (pore opening-76.2 μm , wire diameter-50.8 μm) is the mesh with the smallest pore opening that is available from the manufacturer. Using this mesh, approximately 80 μm diameter droplets can be printed. In order to eject out smaller drops, we need mesh with smaller pore sizes. Copper was electroplated on mesh type #0.0020 to reduce

the pore size. We were able to reduce the pore opening to $\sim 32 \mu\text{m}$. After electroplating the wire diameter increased to $\sim 94.7 \mu\text{m}$ (Figure R2a).

Figure R2: Electroplated mesh characterization. (a) SEM images of the electroplated mesh (pore opening- $32.1 \mu\text{m}$, wire diameter- $94.7 \mu\text{m}$), (b) SEM images of etched superhydrophobic electroplated mesh (pore opening- $25.2 \mu\text{m}$, wire diameter- $101.2 \mu\text{m}$). Contact angle on (c) electroplated mesh and (d) superhydrophobic etched electroplated mesh. (e) Time lapse photographs showing the ejection of smallest droplet (diameter $\sim 42 \mu\text{m}$) using superhydrophobic electroplated mesh.

The copper electroplating process schematic and parameters are shown in Figure R3. The electroplated mesh was processed like other meshes to obtain superhydrophobic pores. SEM images of the electroplated and etched mesh are shown in Figure R2b. The contact angle on as electroplated mesh and etched superhydrophobic mesh was found to be $92^\circ \pm 2^\circ$ and $161^\circ \pm 4^\circ$ respectively (Figure R2c,d). Drop impact printing experiments were performed on the

electroplated superhydrophobic mesh and a drop size of $\sim 42 \mu\text{m}$ in diameter was obtained (Figure R2e). Supplementary Video S6 shows the $\sim 42 \mu\text{m}$ droplet printing. This approach uses a simple electroplating process to reduce the pore opening of mesh for printing droplets of smaller size.

Figure R3: Schematic of the electroplating process.

Scalability: The technique presented in this work can print droplets sizing from $42 \mu\text{m}$ to $960 \mu\text{m}$ by appropriately selecting the mesh size. Such a large scalability in the size of dispensed droplets is better than most technologies. This gives us the versatility to quickly change between filling larger areas and printing fine linewidths by quickly changing the mesh. Quick change of the dispensed droplet size is not easily possible in most conventional printers.

Limitation in terms of mass loading leads to limited thickness of a single printing step. To overcome this limitation and obtain thicker lines, most printing techniques use multiple print passes. In drop impact printing, this is solved by printing droplets with high mass loading. We

could print $46 \mu\text{m} \pm 1.4 \mu\text{m}$ thick line with droplet size of $155 \mu\text{m}$ using ZrO_2 ink with 71% mass loading. This allows us to print thick features in a single pass or single printing step.

Further, a single mesh supports multiple impact locations which can be simultaneously used for printing multiple droplets. Thus, the printing process can be parallelised. This has been shown in Supplementary Video S8. The current printing process shows scalability in terms of feature size, deposition thickness and parallel dispensing.

3D Printing: The 3D printing capability has been shown by printing pillars of different dimensions using different sieves. ZrO_2 nanoparticles (44.4% (w/w)) dispersed in a mixture of Ethylene Glycol and water in the ratio 1:10 was used as an ink for printing. The substrate on which printing was performed was a Teflon coated glass slide kept above a heater at temperature $70 \text{ }^\circ\text{C}$. Figure R4 shows the 3D printed pillars of different dimensions using different sieves. 3D printing is a complex process, which depends on

- i. capabilities of the printing technology,
- ii. properties of the ink, and
- iii. printing conditions (e.g. stage temperature, dispensing frequency, etc.)

Demonstration of successful 3D printing requires careful and tedious optimization of all the above. Even though we have demonstrated 3D printing, ink and the printing conditions can be further optimized to obtain better features.

Figure R4: 3D Printed pillars. (a) using mesh #0.012, (b) using mesh #0.009 and (c) using mesh #0.0045.

- The authors claim to have introduced a new “3D printing” technique. While I generally believe that the process could be suitable for 3D applications, there are no such demonstrations and I have concerns that the process, in its current state, is not suitable for 3D printing. This is due to the positioning accuracy that is low compared to the droplet size (i.e., the average deviation from the targeted position is higher than the droplet size). I suggest removing all such claims or demonstrate actual 3D structures (with the same feature size and resolution as claimed).

Response: 3D printing capability has been shown above. We could achieve this through studying the printing accuracy of the drop impact printing technique. In the original submission, we had reported that the droplets are often ejected with an angle with respect to the vertical axis. In our current setup, the impinging droplet is not aligned to the mesh pore (Figure R5a,b). Absence of concentricity imparts a horizontal velocity component to the ejected droplet. This leads to the observation of ejection at an angle. As the current setup does not have a provision for aligning the mesh and the syringe, we observe a different angle each time the mesh is repositioned with respect to the syringe. However, if the mesh is not moved, the impact process continues to eject microdroplets with the same angle (Video R1). This helps us in obtaining sufficient accuracy for 3D printing when the substrate is kept at a distance of 1.5 mm from the mesh (#0.012, #0.009 and #0.0045). Positioning error was estimated by

printing multiple drops. Figure R5c,d plots the deviation in the position while we attempted to print droplets along a straight line. The worst-case deviation was $\sim 30 \mu\text{m}$ in the lateral direction and $\sim 10 \mu\text{m}$ in the longitudinal direction for mesh #0.012 and #0.0045 respectively. It is interesting to note that this is significantly smaller than the size of the droplets that can be printed with the respective meshes (Figure 1(d) main manuscript). Therefore, 3D printing is possible by keeping the mesh fixed with respect to the syringe while dispensing multiple droplets. Even though our current setup does not incorporate pre-alignment of the impinging drop with the mesh pore, developing such a mechanism is possible using a camera and a motorised stage.

Figure R5: Position accuracy calculation: (a) zero-degree angle ejection and 10-degree angle ejection for mesh #0.012, (b) zero-degree angle ejection and 6-degree angle ejection for mesh #0.0045 and plot between (c) longitudinal deviation (standard deviation) versus ejecting angle for mesh #0.012 and #0.0045 (d) lateral deviation (standard deviation) versus ejecting angle for mesh #0.012 and #0.0045.

- The presented process produces a lot of excess material, as only a fraction of the initially (large) droplet is eventually deposited onto the surface. I could not find the exact number in the manuscript but would estimate it to be as high as 1/1'000 of the volume. Even if this material can be recycled after the printing process (which only works for few inks), it renders the process highly inefficient, if not unsuitable, for numerous inks and applications, many of which the authors claim to be “printable”, such as cell-laden inks. It is simply infeasible (both monetarily and from a synthesis perspective) to produce 1'000x the amount of cell-laden ink that is needed for the final part.

Response: The ratio of the ejected volume to the impact volume for drop impact printing technique is small as seen in Table RT1. We have proposed in the manuscript that the repelled droplets will be collected and can be used again. The process of recycling of inks is very common in continuous inkjet printing, where the undesired droplets those are not suitable for printing are recycled back to use for printing again^{1,2}.

Mesh Type	Mother Drop Volume, V_M (nL)	Ejected Drop Volume, V_e (nL)	V_e/V_M
#0.0020	8685.4	0.280959356	3.2E-5
#0.0045	8685.4	3.123229732	3.6E-4
#0.0009	8685.4	19.16847848	2.2E-3
#0.012	8685.4	363.9018759	4.2E-2

Table RT1: Showing volume of mother drop, ejected drop and their ratio for different meshes.

We agree with the reviewer about the fact that certain bio-based inks having very short decay times³ are not suitable for recycling. For those inks, we have also proposed a method of printing DMEM drops onto a substrate. Upon drying the DMEM forms a hydrophilic patch. Then swiping cell laden solution over the surface leads to generation of smaller cell laden droplets on top of those hydrophilic patches (Figure 4d main manuscript). This way of printing can be used for costly inks and quickly decaying bio-based inks. Using this approach, the quantity required for printing non-recyclable inks can be significantly reduced. For swiping of an area

(6.5 mm x 5.3 mm) and covering 12 number of droplets having base diameter 930 μm , we need 5 μL volume of ink ($\text{Volume}_{\text{trapped}}/\text{Volume}_{\text{swiped}}=0.64$). This value can further be improved (volume of ink required will be less) based on droplet size (smaller diameter droplets) and dots per inch (DPI) requirements.

- The materials palette is not as broad as claimed by the authors, when compared to processes other than material jetting. In fact, it is much smaller than that of, e.g., acoustophoretic printing.

Response: Compared to the acoustophoretic printing technology, we agree with the reviewer that the present material palette is not so broad. In the manuscript we have shown printing aqueous suspensions of nanoparticles, glycerol solutions of various viscosities, cell laden solutions, PEG (non-Newtonian fluid), Polyacrylic acid (PAA) and Xanthum gum (viscoelastic fluid). However, printing high viscosity liquids (viscosity $> 33\text{mPas}$) is still a challenge. As viscous effects increase, capillary waves essential for formation of the cavity are suppressed. Hence, obtaining a single drop through recoil ejection becomes difficult for highly viscous liquids. However, such limitations could possibly be overcome through further modifications of the surface and the technique. These modifications have however not been pursued in this work.

Drop impact printing could work with liquids having surface tension within the range of 72mN/m to 32mN/m. This range is similar with that of inkjet printers⁴⁻⁶. Printing of higher surface tension liquids (liquid metals, alloys etc.) should be possible. However, it has not been attempted due to their toxic nature and difficulties in handling them. Capability to print lower surface tension liquids is however, limited by the failure of the superhydrophobic surface during impact. Superomniphobic surface is a possible option to extend the capability of the current technique to handle lower surface tension liquids.

- One of the claimed, major advantages is the printing of inks with high mass loadings. However, some of the mass is (stochastically?) sieved out by the mesh, which a) clogs the sieve (and thereby diminishes one of the acclaimed advantages over other processes), b) reduces the effective mass loading, and c) reduces the repeatability.

Response:

Clogging of the sieve

The ability to print higher mass loading without clogging is big advantage of this technique. The main reason of clogging in inkjet printing technology is the evaporation of the meniscus that leads to accumulation of nanoparticles at the nozzle periphery due to continuous contact of the ink with the nozzle surface (Figure R6a). In drop impact printing, the ink and mesh pore (which acts as the nozzle) contact only for 8.6 ms. Short contact time eliminates any chance of nanoparticle settling and clogging of the mesh pore due to solvent evaporation (Figure R6b). Therefore, this technique provides us way to have clog free printing.

Figure R6: (a,b) Clogging and anti-clogging mechanism for inkjet nozzle and mesh respectively. (c) Time lapse photography of 71.1% ZrO_2 nanoparticle suspension ejecting out single drop and contact time was estimated to be approximately 8.6 ms. (d) Digital image of printed drops. (e) SEM image of 71.1% ZrO_2 nanoparticle suspension deposited drop and their magnified view and (f) Optical profilometry image of 71.1% ZrO_2 nanoparticle suspension deposited drop for three meshes (#0.012, #0.009 and #0.0045) used for height measurements (Scale bar -100 μ m).

For the current technique, authors agree that accumulation of nanoparticle residues takes place on the mesh after multiple impacts (see Figure R1). These accumulations impede the printing

process by modifying the mesh wettability. However, such accumulation can be easily removed by a mild jet of water followed by drying in N₂ as shown in Figure R1 below.

Figure R1: Sieve contamination characterization. (a) pinned nanoparticles on superhydrophobic sieve (#0.012) after approximately 1000 drop impacts. (b) cleaning process of sieve after contamination (water jet impact and N₂ purging). (c) Mesh after cleaning. (d) Static contact angle of water droplet on cleaned sieve and (e) Water droplet repellence test by impacting drops on superhydrophobic cleaned mesh.

Effective mass loading

Reviewer has further asked about effective mass loading of the dispensed droplet. In order to address this, we have impacted 50 drops of ZrO₂ nanoparticles (concentration: 71.1%) suspended in ethylene glycol to water in the ratio 1:10 on the mesh #0.012, #0.009 and #0.0045 (Figure R6c). The drops were collected on Teflon coated glass slides. After deposition the droplets were dried (Figure R6d,e). Height of the deposited material (after drying) was measured using optical profilometry (Figure R6f). The effective mass loading (%) was estimated by calculating the deposited mass and the total volume of the ejected droplets. The average drop volume was calculated from the droplet diameter measured using ImageJ from the high-speed imaging of the printed droplet while it is in air (Figure R6c). Average height of

the deposited droplet and average diameter of the deposited droplet are shown in Table RT2 for three sieves #0.012, #0.009 and #0.0045. From these values, the effective mass loading of the printed drops was found to be approximately ~67% for all three meshes. The drop in the small mass loading percent could be due to some level of scaling on the pipeline and syringe during flowing of the ink from the reservoir to syringe.

In the main manuscript, the reported drop height is 16.9 μm for #0.009 mesh using 71% mass loading ink (Figure 3e main manuscript). The reported drop height in the main manuscript is less as compared to Table RT2 because the droplet was printed on a plain glass substrate without Teflon coating, which is hydrophilic, and hence the droplet spreading was more on the hydrophilic substrate resulting in reduced height. The data presented in Table RT2 is on glass substrates coated with Teflon and hence the droplet spreading was less on the Teflon coated substrates resulting in increased heights of the printed droplets. For 71% mass loading ink, the effective mass loading for the printed droplet in Figure 3e (main manuscript) was found to be 66%. From this data it is clear that the reduction in mass loading of the printed droplets using drop impact printing is very less (~4%).

Sieve type	#0.012	#0.009	#0.0045
Ejected drop diameter (measured using ImageJ software)	807 $\mu\text{m} \pm 11 \mu\text{m}$	408 $\mu\text{m} \pm 12 \mu\text{m}$	155 $\mu\text{m} \pm 6\mu\text{m}$
Deposited diameter (measured using ImageJ software)	696 $\mu\text{m} \pm 25 \mu\text{m}$	383 $\mu\text{m} \pm 10\mu\text{m}$	151 $\mu\text{m} \pm 7\mu\text{m}$
Deposited height (measured using optical profilometry)	217 $\mu\text{m} \pm 4 \mu\text{m}$	97 $\mu\text{m} \pm 2 \mu\text{m}$	46 $\mu\text{m} \pm 1.4 \mu\text{m}$
Effective mass loading (%)	66% $\pm 1.1\%$	67% $\pm 3.2\%$	67% $\pm 1.46\%$

Table RT2: Parametric values of droplet diameter in air after ejection, drop diameter after deposition, drop height after deposition and effective mass loading after deposition for all three meshes (#0.0045, #0.009 and #0.012). The droplets were collected on Teflon coated glass slides.

Repeatability

The repeatability of printing drops using high mass loading (44.4%) suspension solution is shown in Figure R1. Approximately after 1000 impacts, there was some degree of contamination on the spot where drop impacted the mesh as seen in Figure R1a. These contaminations were easily removed by washing with a mild jet of deionized (DI) water followed by nitrogen dry. Thus, printing high mass loading inks doesn't limit the process capabilities.

I, thereby, recommend major revisions with an emphasis on proving the claims in question quantitatively or with specific references; or removing these claims (which, however, reduces the novelty and impact of the work).

In detail:

- The authors state that their process “outperforms conventional inkjet technique in most aspects.”. Besides the investigated factors, could the authors comment on other, important, metrics, such as the resolution (I believe inkjet printing can go below 10 μm , whereas the current process seems to be limited to 70 μm), throughput volume per time per nozzle (in comparison to inkjet printing), the scalability (inkjet printing can be scaled to hundreds of nozzles with minimal distance between the nozzles), and the positional accuracy?

Response:

Resolution: Using the process described in response to a previous question we have demonstrated that the present technique can now print droplets of $\sim 42 \mu\text{m}$ in diameter. Further reduction in droplet size is possible through optimization of the fabrication process discussed above. With this improvement the presented technique becomes competitive with most commercially available material jetting systems.

Throughput volume per time per nozzle: In terms of number of droplets, we have been able to achieve 6 drops per seconds. This is slower compared to several other techniques (e.g. Acoustophoretic printing, EHD printing, etc.). However, we can achieve very high throughput in terms of dispensed volumes. This is due to the technique's ability to print droplets with a significantly wider range of diameters (42 μm – 960 μm). Further individual mesh supports multiple simultaneous droplet impacts providing parallelization.

Scalability: This technique can also have multiple nozzles as shown in Supplementary Video S8. Each impinging droplet has a diameter of ~ 2.5 mm. This implies that we can have a system of 100 parallel nozzles in an area of 10 cm x 10 cm (assuming a pitch of 1 cm between the droplets). In comparison, microfabricated devices with similar number of nozzles will occupy smaller area. However, as discussed above we can achieve much higher volume throughput due to: (i) capability to dispense higher mass loading droplets; and (ii) capability to dispense a wider range of droplet diameters.

Position accuracy: Position accuracy of drop impact printing has been discussed in detail above. If the position of the mesh and the syringe are kept fixed with respect to each other, this technique achieves high positional accuracy with less than ~ 30 μm deviation for the worst-case scenario (printed drop size 860 μm) as shown in Figure R5 above which is significantly less as compared to the size of the printed drop.

- Lots of questionable statements are made without proof. E.g., “However, newer applications require printing of more exotic inks”: What are newer applications (as compared to which “old” applications)? And what are more exotic inks (as compared to “less” exotic inks)? This is just one example, I recommend carefully reviewing all such statements throughout the manuscript.

Response: We have reviewed the manuscript and modified such statements.

- The authors state that inkjet printing (more accurately and commonly referred to as material jetting) “fails to provide the resolution, accuracy and widespread applicability”. To my understanding, material jetting provides higher resolutions and higher accuracies as the process proposed.

Response: We agree with the reviewer that some material jetting techniques provide high resolution and accuracies. However, our statement was with respect to the specific inks which are of current research interest (e.g. biological samples, high mass loading ink for 3D printing, etc.). For conventional material jetting techniques, ink concentration ranges from 15% to 30% (wt%). Suspended particle sizes are limited to few microns^{7–10}. Detail comparison of printing technologies is presented in the main text. We have modified some statements in the manuscript to provide better clarity of our technique’s objective and novelty.

- In turn, the authors claim that alternative printing techniques that use, e.g., acoustic forces or microfluidics, to overcome these aforementioned “limitations” are “restricted to a selected community”. I strongly disagree with this statement and believe that some of these techniques have incredible (and even higher) potential. For example, acoustophoretic printing can eject (“mass loaded”) droplets over significantly larger viscosity ranges than any other process, including the process proposed.

Response: We agree with the reviewer that some of these techniques have very high potential. The purpose of the statement was not to compare technical abilities of different techniques. The statement tries to highlight the simplicity of the proposed setup. Nozzle for microfluidics-based droplet generation techniques requires microfabrication facility. These techniques

require highly sophisticated tools and indirectly increases the cost of the technique. Additionally, such facilities are not readily available in many countries. Other printing techniques use nozzles with integrated actuators, which are often driven by a complex drive / control system.

Using sieve for printing doesn't require the use of sophisticated tools or facilities. We believe that such a system will be more readily available to labs with very limited resources. Use of the phrase "restricted to a select community" did not imply that these techniques are of limited use. The intention was to indicate restrictions in accessibility of expensive printing systems in labs with limited resources. The statement marked by the reviewer has now been rewritten in the manuscript.

- What do the authors mean by nozzles with "complicated actuators"?

Response: Dispensing droplets smaller than capillary length requires use of an external force. For example, thermal and piezoelectric actuation are among the most common ones. For nozzles using piezoelectric actuation, the piezoelectric actuators are mostly fabricated on the glass nozzle. Integration of the piezoelectric actuators on the nozzle makes their fabrication process complex. To optimize for printing of various liquids, these actuators are driven using complex waveform actuation¹²⁻¹⁴ requiring dedicated drive electronics. Further, integration of pressure control is also required to hold the liquid meniscus at the nozzle tip. Integration of these control and actuation mechanisms makes the system complicated and increases the cost of the set up. The statement in the main text has been changed to "with integrated actuators and a complex drive / control system".

- The authors claim that other techniques are limited because of clogging of “expensive nozzles”. To my understanding, “cheap” nozzles are commonly and commercially available down to (inner) diameter sizes of 10 μm or smaller (which is smaller than what the current process can produce). I do not agree with this argument.

Response: Printheads used for printing dyes on paper are available at low-cost (~ \$50). These printheads are however useful for printing very specific inks with a very limited range of material properties. Such printheads are never used for printing a wide range of materials. To the best of our knowledge, printheads which are suitable for printing a wide variety of inks are available in the price range of hundred to thousand dollars. Comparing to this, for drop impact printing technique, the fabricated mesh (3 cm x 2cm) costs approximately \$9.4.

- The authors state a whole list of issue with all existing technologies and claim that their technology overcomes “these issues”, which leaves the impression that their technique is better than each and every of the existing techniques. I believe that their technique has some advantages, but also disadvantages over existing technologies and recommend reviewing all such statements.

Response: We thank the reviewer for bringing up the ambiguous statement. We modified these statements to be more specific in terms of advantages and disadvantages of the presented technique.

- The same goes for statements like “[...] outperforms conventional inkjet technique in most aspects.” and others. Such statements are highly vague. Please also provide specific proof on all such statements.

Response: The statements that the reviewer marked above have been modified to be more specific in terms of the technique's advantages and disadvantages. The few silent features of our technique are

1. Capability to print droplets with high mass loading.
2. Capability to print large particles (i.e. comparable to the dispensed droplet diameter).
3. Large range of printed droplet volumes (diameter varying from $\sim 42 \mu\text{m}$ to $\sim 960 \mu\text{m}$).
4. Satellite free printing.
5. Simple setup.
6. Low operational cost.

These all are important aspects in printing technology and are resolved using drop impact printing technique. Using nozzle free approach, we found that this technique can solve many problems that recent conventional printers face. However, we do agree with the reviewer that in some respects like printing higher viscous liquid is still a challenge. Present literature gives us a way to print viscous liquids using acoustophoretic printing technology¹¹. Even though the printing accuracy of our technique is sufficient for 3D printing, it is still not amongst the best. We have tried to compare all these aspects and cited papers those are relevant in the revised manuscript.

• The authors claim that their process is unique in terms of the mass loading it can achieve (71%). Can the authors state what the (theoretical and practical) maximum values of (all of) the other processes are, they compare their process to?

Response: Existing literature have reported printing with mass loading concentration up to 40%. Some of the literature that have shown high mass loading are shown in Table RT3 below. Theoretical limit of mass loading for any printing technique is determined by the surface

tension and viscoelastic properties of the drop. As mass loading increases the effective viscosity will increase. Further addition of stabilizing agents often reduces the surface tension. However, it is the practical limit due to clogging which prevents nozzle-based techniques to perform better. As explained above evaporation led nanoparticle accumulation at the nozzle tips leads to clogging. Thus, tendency to clog increases with decrease in nozzle tip diameter and increase in the mass loading.

Literature	Mass loading	Application
T. Wang et.al., 2005 ¹⁵	40%	PZT using inkjet printing
M. Shlomo ⁴	5-30%	Printed Electronics
B. Derby, 2015 ¹⁶	20% - 30%	Ceramics Components
J. Sadie, 2015 ¹⁷	40%	Metal Nanoparticles printing

Table RT3: Literatures reporting high concentration inks in terms of their mass loading (%).

- How do the authors generate the initial droplet? Please provide a detailed description.

Response: The liquid is pumped from reservoir to syringe at the top. The initial drop (mother drop) is generated using syringe kept at a height. The schematic Figure R7 shows the details of the drop impact printing setup (It has also been added to the revised supplementary information file).

Figure R7: Schematic of drop impact printing setup with proper detailing of accessories.

• The authors say that their process is “cost-effective, compact in size”, when compared to existing methods. Can the authors quantify these statements, e.g., what is the size and cost of the complete set-up, and compare it to the other, competing methods they refer to? I believe that this is a very bold statement, which needs to be put into relation quantitatively. Can the authors add the droplet generation hardware to the schematic (and possibly add a photo)? Is the droplet generation hardware part of this “cost-effective, compact in size”-evaluation (which it should be)?

Response: The drop impact printing setup is shown in Figure R7. It has two major units. The “ink pumping unit” which pumps the liquid ink from the reservoir to the syringe. The syringe is positioned at the dispensing height using a stage. The mesh is positioned on a two-axis stage and the substrate is placed on a three-axis stage. The setup dimension (including ink reservoir, z-stage, substrate and mesh stage) is around 12 cm x 10 cm x 10 cm. The system cost will include the automated stages, the pump unit and reservoir. From the point of view of the

printing technique, the operational cost is only the cost of the mesh (nozzle in other techniques). The cost for a mesh of size 3 cm x 2 cm including fabrication is estimated to be around \$ 9.4 (USD). In comparison, microfabricated printheads capable of jetting multiple materials cost around \$100. Printheads fabricated using glass nozzles which are again capable of jetting multiple materials cost significantly higher (> \$500). Even when compared to printheads optimised for printing dyes, operational cost of the current technique is significantly lower.

When dispensing nanoparticle laden ink, nozzle clogging is one of the most important problems for all technologies. Drop impact printing is immune to clogging as discussed above. Further mesh replacement is economical compared to other techniques.

It is important to note that fabrication of the mesh is simple enough to be replicated in any lab with access to etching chemicals. The dispensing setup is also simple and can be built without significant difficulties. This technique will allow low resource labs (including school labs) to experiment with droplet printing.

- The sieve is weaved in different, vertical planes (Figure 1e). This creates a strong directionality of the holes or “channels” away from being orthogonal to the plane of the sieve. How does this effect the direction of the generated, small droplet? Are the authors taking this into account when they eject the “big” droplet onto the surface, i.e., do the authors know which hole will be dispensing the “small” droplet?

Response: The directionality of the holes in addition to absence of concentricity leads to ejection of the generated droplet at an angle with respect to the vertical. Figure R5 (explained above) demonstrates the effect of droplet ejection at an angle. However, if the relative position of the impinging droplet is kept fixed with respect to the pore opening, the ejection angle

remains constant. This will provide minimum displacement error. The positioning of mother drop can be easily achieved using a normal camera.

• When looking at the paper that initially discovered the penetration of drops through superhydrophobic meshes (reference 31) and the methods in the presented work, it seems like there is a very narrow range in which (single) droplets can be reliably ejected (“impact cavity mode”). First, I assume it takes significant effort to find this regime for every new material, which stands in contrast to the simplicity of the approach the authors claim? Second, does this not limit the materials palette of significantly (which stands in contrast with the versatility the authors claim)?

Response: The analysis in Figure 2 (main manuscript) shows the printing zone of drop impact printing. The viscosity and surface tension range were shown in the zone. For a particular surface tension and viscosity, the plot (Figure 2c) shows the required height. Based on this data, printing of new liquids for known surface tension and viscosity will be easy. Although some optimisation is required, but that can be easily achieved with the help of prior characterisation shown in Figure 2c. We believe that the material palette will not be limited. Mostly all liquids that are within this surface tension and viscosity range will work for this technique. We also accept that printing of higher viscous liquids like honey is still a challenge. And this is our future goal to achieve. We are looking forward to modifying the technique to print higher viscosity liquids.

• The materials the authors claim to be ejectable differ in more properties than only the “viscosities and surface tensions” the authors investigate. A full rheological analysis is necessary.

Response: The present analysis (Figure 2) shown in the main paper is for Newtonian fluid: glycerol water mixture for viscosity comparison and ethanol water mixture for surface tension. The rheology of these two types of liquid mixtures is already well reported in the literature^{18,19}. For other liquids (nanoparticle-based solutions) suitable references were taken from literature to know its approximate surface tension and viscosities.²⁰⁻²² The values gave us the approximate idea to keep our solutions in printing zone. We agree that other rheological properties will matter. However, we believe that detailed studies of those are not in interest of this paper.

- Can the authors show examples of the ejection of these specific materials (i.e., ceramic and cell-laden inks)? Can the authors show that, in particular, the cells in the cell-laden inks survive the ejection process?

Response: Live HeLa cells printing using mesh #0.009 and mesh #0.012 was performed, and cell viability was estimated. The cell viability was calculated for 10 printed droplets. Figure R8 shows cells containing inside a drop. The cells solution dispersed in PBS was mixed in trypan blue dye in ratio 1:1. The dead cells were stained by trypan blue and appeared blue in colour. The same can be observed in Supplementary Video R2. The cell viability after printing was found to be 86.1% for mesh #0.009 and 94% for mesh #0.012.

Figure R8: Printed droplet containing HeLa cells. The red circled cells are dead cells that appear blue in colour due to trypan blue staining.

• What happens to the excess material? What is the ratio of the volume of the initial droplet to the ejected droplet?

Response: The excess material is repelled back and goes to the reservoir to be used for printing again. The ratio of the volume of the initial droplet to the ejected droplet is listed in the Table RT1.

• The presented drop impact printing covers materials with Z numbers ranging from 2 to 200 (Figure 2d). This does not outperform inkjet printing on the lower Z number regime. Further, acoustophoretic printing, for example, covers materials with Z numbers ranging from 10^{-3} to 10^3 , which is significantly higher. Could the authors generate a more complete comparison in Figure 2e that shows not just inkjet and drop impact printing, but also all the other methods the authors claim are outperformed with their approach?

Response: Figure 2e is modified and more detailed comparison with other techniques has been shown.

• In the manuscript, the authors say that “Drop-impact printer can print for Z values varying from 3 to 200”, but Figure 2d shows a range of 2 to 200. Please correct.

Response: Apologies for the mistake. Same has been corrected.

• Figure 3a: Shouldn't the green bar go down to 10^2 rather than the 10^3 , where it is currently (based on the text “100 times greater than the particle size”)? How do these values compare to the other methods mentioned in the introduction of the manuscript? Could the authors add bars for these methods (or, alternatively, remove the claim that they outperform these methods)?

Response: The green bar is corrected in the main text and the L/D_p ratio for new techniques (that are available in literature) have been added.

• The authors say that they can eject particles with diameters of up to $1/4$ of the sieve pores. The authors further claim that they can eject inks with mass loadings of up to 71% (Figure 3). Can the authors quantify the particle loading (or mass) ratio of the ink before and after extrusion?

Response: We found the mass loading percentage of ejected drop was approximately 67% for ink that have 71.1% mass loading (Details of experiment in Figure R6, Table RT2). There is some drop in mass loading percent and it can be due to some level of scaling on the pipeline and syringe during flowing of ink from reservoir to syringe.

• Further, can the authors quantify the particle loading (or mass) ratio of a significant number of ejected drops to show that it is consistent (and that the sieve doesn't filter the particles or prevent them from ejection)?

Response: The effective mass loading percent was estimated by performing experiments for 50 drops on mesh #0.012. Figure R6 and Table RT2 above shows the detailed study.

• Clogging is often a function of time. Can the authors quantitatively show that they can maintain ejection of the high mass loading inks through the same pore over an extended amount of time (to maintain comparability with existing, nozzle-based processes)?

Response: We agree with the reviewer statement. The multiple ejection of high mass loaded nanoparticle solution (44.4%) through a single pore (mesh type #0.012) was investigated. To explain this, we performed an experiment where we impacted approximately 1000 drops on mesh (#0.012). After 1000 impacts, we observed some degree of contamination on the spot where drop impacted the mesh (Figure R1a). However, the contamination can be easily removed with a mild water jet washing followed by N₂ purging (Figure R1b,c). In order to quantify the wettability, we also measured the contact angle and contact angle hysteresis (Figure R1d). The contact angle was $157^{\circ} \pm 3^{\circ}$ and hysteresis was $< 5^{\circ}$ (contact angle hysteresis was calculated from the advancing and receding angle of the droplet at the onset when the droplet starts to slide) satisfying the conditions of superhydrophobicity. Hence, authors believe that although there will be some sieve contamination, the same can be resolved with a mild water wash and clogging of the mesh pore can be completely minimized.

Figure R1: Sieve contamination characterization. (a) pinned nanoparticles on superhydrophobic sieve (#0.012) after approximately 1000 drop impacts. (b) cleaning process of sieve after contamination (water jet impact and N₂ purging). (c) Mesh after cleaning. (d) Static contact angle of water droplet on cleaned sieve and (e) Water droplet repellence test by impacting drops on superhydrophobic cleaned mesh.

• The authors state “Unlike clogged nozzle, for our case the residues can be easily removed by washing with a secondary liquid.”. Do the authors mean that this would happen during the printing process? Also, I assume that this strongly depends on the ejected materials. Many materials might cure onto the sieve and form a strong bond that cannot easily be washed away.

Response: After several impact on the mesh, we found that the ink (nanoparticles-based suspensions) was pinning on the mesh. The above statement is written in order to explain how to remove the pinned nanoparticles from the mesh after the experiment. We found that simply by water jet impact and nitrogen dry, the pinned nanoparticle can be easily removed (Figure R1). We agree with the reviewer that some inks might not be amenable to cleaning approach. The statement in the manuscript is remodified to have more clarity.

• A droplet displacement error of 90 μm at a print height of 1 mm is significant, when the droplet diameter is 82 μm , and contradicts the “high accuracy” claim. Can the authors compare this number to other, existing processes (including inkjet printing and other techniques)?

Response: The position accuracy calculation previously presented in the manuscript was based on the angle of deviation in droplet ejection. When the sieve is moved, the concentricity of the impacting drop with respect to pore also shifts. Due to which the microdroplet is ejected with varying angles and hence, a large deviation in drop placement was observed. This represents the worst-case scenario in the absence of any alignment.

However, if the relative position of the mesh and the syringe are kept constant, positional deviation can be minimised. We found that if the ejected drop (155 μm in diameter) is

concentric with the mesh (data shown for mesh type #0.0045), then the position accuracy can be as high as 8 μm (longitudinal) and 6 μm (lateral) (substrate kept at distance of 1.5 mm from the mesh). The other printing techniques' accuracy is listed in Table RT4 below. This shows the ability of the drop impact printing to print droplets with comparatively high accuracy.

Printing Technique	Nozzle distance, mm	substrate	Printed Drop size, μm	Accuracy, μm
Drop On Demand Inkjet ²³	1		~ 120	66
EHD Printing ²⁴	1		~ 8	10
Acoustophoretic printing ¹¹	3.15		~ 100-500	60

Table RT4: The comparison of longitudinal position accuracy (standard deviation) for different printing techniques.

- Please review the grammar and style throughout the manuscript.

Response: We have reviewed and wherever necessary corrected the grammar and style throughout the manuscript.

Reviewer #3 (Remarks to the Author):

In this manuscript, the authors have proposed a new method for printing using droplet impact and a hydrophobic sieve. The authors showed that this approach could address the drawbacks of inject printing. I should assess this article in two aspect: one on novelty of this method and its promise and one on the fundamental aspect of this work.

The authors rationalize that formation of satellite droplets and clogging are two drawbacks of inject printing approach and this new method could address these drawbacks. I would encourage the authors to think more carefully before making this claim.

In this approach, the generated droplet has volume of 100 times less than the impact droplet. What will happen to the rest of the droplet? It will be all repelled and should be somehow recycled. This is much worse than any inject printing approach in which the droplet sizes and flow rate could be accurately controlled.

The second drawback of the inject printing is clogging. The clogging problem will be much worse in the sieve compared to a short nozzle in inject printing. By appropriate surface coating of the nozzle, this problem could be addressed. Thus, this new proposed method does not offer any advantage over inject printing and in some aspects is worse.

On the fundamental aspect of this proposed approach, the authors have completely neglected the hammer pressure developed at stagnation point. This pressure could be few orders of magnitude higher than dynamic pressure. There is no new fundamental understanding in this work.

I would suggest the authors to correct the fundamental argument in this work and just focus on their observation rather than claiming its superiority over inject printing.

Response:

We would like to thank the reviewer for the valuable inputs. This will definitely help improve the quality of the manuscript. We have updated the manuscript to focus on the current observations and to be more specific in terms of the technique's advantages and disadvantages.

The few silent features of the presented technique are

1. Capability to print droplets with high mass loading.
2. Capability to print large particles (i.e. comparable to the dispensed droplet diameter).
3. Large range of printed droplet volumes (diameter varying from $\sim 42 \mu\text{m}$ to $\sim 960 \mu\text{m}$).
4. Satellite free printing.
5. Capability to perform 3D printing
6. Simple setup.
7. Low operational cost.

There are three major concerns from the reviewer. We have tried to address each point with experimental results and calculation.

(1) We agree with the reviewer that the ratio of the ejected volume to the impact volume is less than 100 times. We have proposed in the manuscript that the repelled droplets will be collected and can be used again. The process of recycling of inks are very common in continuous inkjet printing, where the undesired droplets those are not suitable for printing are recycled back and used for printing again^{1,2}.

(2) In case of inkjet printing, clogging takes place at the nozzle due to liquid evaporation leaving particles on the inner periphery of the nozzle. The liquid is always in constant contact with the nozzle in case of inkjet printing. In contrast, in case of drop impact printing, there is

no continuous contact between the liquid to be printed and the mesh except for the time of impact (contact time is approximately 8.6 ms). Hence this reduces the clogging probability. In order to examine that, we performed drop impact of ZrO_2 nanoparticles dispersed in a mixture of Ethylene Glycol and water in the ratio 1:10. The drop impact of this solution was carried out for approximately 1000 times at one place using the mesh #0.012 coated with Teflon instead of stearic acid (Please refer Table S1 for the geometrical parameters of the mesh). After 1000 impacts, we observed some degree of contamination on the spot where drop impacted the mesh (Figure R1a). However, the contamination can be easily removed with a mild water jet washing followed by N_2 purging (Figure R1b). In order to quantify the wettability, we also measured the contact angle and contact angle hysteresis (Figure R1c). The contact angle was $157^\circ \pm 3^\circ$ and hysteresis was $< 5^\circ$ (contact angle hysteresis was calculated from the advancing and receding angle of the droplet at the onset when the droplet starts to slide) satisfying the conditions of superhydrophobicity. Hence, authors believe that although there will be some sieve contamination, the same can be resolved with a mild water wash and clogging of the mesh pore can be completely minimized.

Figure R1: Sieve contamination characterization. (a) pinned nanoparticles on superhydrophobic sieve (#0.012) after approximately 1000 drop impacts. (b) cleaning process of sieve after contamination (water jet impact and N₂ purging). (c) Mesh after cleaning. (d) Static contact angle of water droplet on cleaned sieve and (e) Water droplet repellence test by impacting drops on superhydrophobic cleaned mesh.

(3) In case of inkjet printing, the resolution is 600 dpi which means printed droplet diameter is ~30-40 μm . However, inkjet printing cannot print droplets with larger diameters. In case of drop impact printing, we can print droplets having a range of diameters (42 μm - 960 μm). This is an advantage of our technique over inkjet printing. Another advantage is the low-cost of our technique. This will allow the use of this technique by a larger community of the society. The commercially available inkjet printer range from few hundred to thousand dollars. The cheap printers in this domain are available that can also be used for material printing purpose but need further modification that indirectly increases the cost²⁵. Comparing to this, for drop impact technique the fabricated mesh which acts as a nozzle, costs approximately \$ 9.4. We have provided here the price of some commonly used printheads.

Printheads	Price (in cost)
Canon PIXUS iP6000D	\$ 34.80
Canon PF-04 iPF650	\$ 468.00
HP 7510	\$ 39.59
Epson XP600	\$ 135.50
Drop Impact Printing	\$ 9.4

(4) Authors appreciate the reviewer's comment on considering the water hammer pressure developed at the stagnation point. We considered mesh #0.009 to answer this question.

The water hammer effect and the dynamic pressure have been found to follow the relationship

$$P_{WH} = aP_D^{26}$$

where a is a scaling pre-factor. Xu et al.²⁷ have found that a is function of the number of pores

N ($N = \frac{\pi D^2}{4(w+L)^2}$, where D =mother droplet diameter, w =wire diameter, L =pore opening)

covered by the droplet during the impact and experiments have shown that $a \geq 1$ for $N = O(100)$.

In our case, we calculated N for mesh #0.009 ($D=2.5$ mm, $w=228.6 \mu\text{m}$, $L=279 \mu\text{m}$).

$$N = \frac{\pi D^2}{4(w+L)^2} = \frac{\pi 2.5^2 \times 10^6}{4(228.6+279)^2} = 19.81$$

$$a \text{ and } N \text{ are correlated as: } a = \frac{8}{1.1 + \frac{466}{N}} - 1$$

N is of the order of 10 in our case. Therefore, the water hammer pressure is expected to be negligible as compared to the dynamic pressure²⁶.

In our case, the surface is superhydrophobic having micro and nanostructures. As we are using mesh, the solid fraction will be very less as compared to flat surface. Dash et.al.²⁸, have reported that for superhydrophobic surfaces, hammer pressure will be two order magnitude less compared to dynamic pressure. This dependence of the water hammer pressure coefficient on the capillary pressure may be explained considering the morphology of the superhydrophobic surface, which is a combination of solid surfaces and air gaps. When a droplet impinges on a flat surface, its motion in the direction of fall is immediately arrested, resulting in a shock pressure. However, in the case of structured surfaces, the droplet experiences a heterogeneous impact. While the droplet comes to a sudden stop on the solid parts of the surface, it is still free to deform into the air gaps so that its overall deceleration is gradual. The shock developed is thus alleviated compared to that of a flat surface.²⁸

Based on this, authors believe that on the superhydrophobic mesh during impact, the hammer pressure can be neglected.

References

1. Basaran, O. A., Gao, H. & Bhat, P. P. Nonstandard inkjets. *Annu. Rev. Fluid Mech.* **45**, 85–113 (2013).
2. Castrejon-Pita, J. R. *et al.* Future, opportunities and challenges of inkjet technologies. *At. sprays* **23**, (2013).
3. Ng, W. L., Yeong, W. Y. & Naing, M. W. Polyvinylpyrrolidone-based bio-ink improves cell viability and homogeneity during drop-on-demand printing. *Materials (Basel)*. **10**, 190 (2017).
4. Shlomo, M. *The chemistry of inkjet inks*. (World Scientific, 2009).
5. Nayak, L., Mohanty, S., Nayak, S. K. & Ramadoss, A. A review on inkjet printing of nanoparticle inks for flexible electronics. *J. Mater. Chem. C* **7**, 8771–8795 (2019).
6. Hoath, S. D. *Fundamentals of inkjet printing: the science of inkjet and droplets*. (John Wiley & Sons, 2016).
7. Reiser, A. *et al.* Multi-metal electrohydrodynamic redox 3D printing at the submicron scale. *Nat. Commun.* **10**, 1853 (2019).
8. Wang, Y. *et al.* Reactive Conductive Ink Capable of In Situ and Rapid Synthesis of Conductive Patterns Suitable for Inkjet Printing. *Molecules* **24**, 3548 (2019).
9. Yu, M., Ahn, K. H. & Lee, S. J. Design optimization of ink in electrohydrodynamic jet printing: Effect of viscoelasticity on the formation of Taylor cone jet. *Mater. Des.* **89**, 109–115 (2016).
10. Wang, J.-C., Chang, M.-W., Ahmad, Z. & Li, J.-S. Fabrication of patterned polymer-antibiotic composite fibers via electrohydrodynamic (EHD) printing. *J. Drug Deliv. Sci. Technol.* **35**, 114–123 (2016).
11. Foresti, D. *et al.* Acoustophoretic printing. *Sci. Adv.* **4**, eaat1659 (2018).

12. Shin, P., Sung, J. & Lee, M. H. Control of droplet formation for low viscosity fluid by double waveforms applied to a piezoelectric inkjet nozzle. *Microelectron. Reliab.* **51**, 797–804 (2011).
13. Wijshoff, H. The dynamics of the piezo inkjet printhead operation. *Phys. Rep.* **491**, 77–177 (2010).
14. Chen, F. *et al.* A piezoelectric drop-on-demand generator for accurate samples in capillary electrophoresis. *Talanta* **107**, 111–117 (2013).
15. Wang, T. & Derby, B. Ink-Jet Printing and Sintering of PZT. *J. Am. Ceram. Soc.* **88**, 2053–2058 (2005).
16. Derby, B. Additive Manufacture of Ceramics Components by Inkjet Printing. *Engineering* **1**, 113–123 (2015).
17. Sadie, J. Three-Dimensional Inkjet-Printed Metal Nanoparticles : Ink and Application Development. (University of California, Berkeley, 2017).
18. Association, G. P. Physical properties of glycerine and its solutions. *New York Glycerine Prod. Assoc.* (1963).
19. Khattab, I. S., Bandarkar, F., Fakhree, M. A. A. & Jouyban, A. Density, viscosity, and surface tension of water+ ethanol mixtures from 293 to 323K. *Korean J. Chem. Eng.* **29**, 812–817 (2012).
20. Kosmala, A., Wright, R., Zhang, Q. & Kirby, P. Synthesis of silver nano particles and fabrication of aqueous Ag inks for inkjet printing. *Mater. Chem. Phys.* **129**, 1075–1080 (2011).
21. Muthamizhi, K., Kalaichelvi, P., Powar, S. T. & Jaishree, R. Investigation and modelling of surface tension of power-law fluids. *RSC Adv.* **4**, 9771–9776 (2014).
22. Wang, X. Drop-on-demand inkjet deposition of complex fluid on textiles. (2008).

23. Park, J. A. *et al.* Freeform micropatterning of living cells into cell culture medium using direct inkjet printing. *Sci. Rep.* **7**, 1–11 (2017).
24. Park, J.-U. *et al.* High-resolution electrohydrodynamic jet printing. *Nat. Mater.* **6**, 782–789 (2007).
25. Cui, X., Breitenkamp, K., Finn, M. G., Lotz, M. & D’Lima, D. D. Direct human cartilage repair using three-dimensional bioprinting technology. *Tissue Eng. Part A* **18**, 1304–1312 (2012).
26. Zhang, G., Quetzeri-Santiago, M. A., Stone, C. A., Botto, L. & Castrejón-Pita, J. R. Droplet impact dynamics on textiles. *Soft Matter* **14**, 8182–8190 (2018).
27. Xu, J., Xie, J., He, X., Cheng, Y. & Liu, Q. Water drop impacts on a single-layer of mesh screen membrane: Effect of water hammer pressure and advancing contact angles. *Exp. Therm. Fluid Sci.* **82**, 83–93 (2017).
28. Dash, S., Alt, M. T. & Garimella, S. V. Hybrid Surface Design for Robust Superhydrophobicity. *Langmuir* **28**, 9606–9615 (2012).

We have made attempts to fully address the points raised by the reviewers. Once again, we wish to express our gratitude to the reviewers for their careful reading of our manuscript, and for their comments and helpful suggestions. We would like to thank the editor for giving us time to address all the points.

We hope the revised manuscript will be acceptable for publication in your highly esteemed journal.

Yours faithfully,

Dr. Prosenjit Sen
Associate Professor
Centre for Nanoscience and Engineering
Indian Institute of Science
Bangalore, India 560012

Reviewers' comments:

Reviewer #1 (Remarks to the Author):

The authors have addressed the issue I had raised in my first review. My suggestion is to publish the paper.

Reviewer #2 (Remarks to the Author):

I appreciate the additional effort the authors put into their work. However, after carefully reviewing the responses, I am (still) not convinced that the claims and, thereby, publication in Nature Communications are justified.

- I believe that 3D printing is possible with the method, but it has not been demonstrated. Pillars of 2-3 droplets are not "3D" and the quality needs to be significantly improved to justify the claim (as opposed to the authors, I do believe that it should be within the scope of this paper).
- The comparison to existing processes is still incomprehensive.
- The authors claim a broad range of printable materials, but the required recyclability of the inks limits the materials palette significantly (e.g., no solvent-based inks, no expensive inks, ...).
- There are too many claims that various things could be possible after further optimization of the technique (including the aforementioned 3D printing or the printing of higher-viscosity inks). I agree that some of these things seem feasible, but others are not so obvious to me.
- The authors claim that clogging, e.g, in Inkjet Printing (again, the only comparison is Inkjet Printing) occurs due to the evaporation of solvents. However, if solvent-based inks are used here, the inks won't be recyclable.
- About 6% of the mass loading gets lost in the process (and likely settles on the sieve?), which I find significant and, depending on the application, intolerable (as opposed to the authors' opinion).
- The mother-drop formation at a height above the sieve, driven by gravitational forces, seems finicky and rather limiting in terms of the throughput speed (the authors mention 6 drops per second). For an actual 3D printing process, more elegant solutions are required. These solutions might be more costly, diminishing their claim to be significantly cheaper than other methods.

Reviewer #3 (Remarks to the Author):

The comments are addressed. The only remaining question is on recycling of the liquid as 99% of the liquid should be recycled on each impact. This is a challenge in real-implementation of this method.

Response to reviewers' comments

Manuscript ID: NCOMMS-19-33513

“Drop Impact Printing”

Reviewer #1 (Remarks to the Author):

The authors have addressed the issue I had raised in my first review. My suggestion is to publish the paper.

We would like to thank the reviewer for providing valuable comments during the review process. This indeed helped improve the quality of the manuscript.

Reviewer #2 (Remarks to the Author):

I appreciate the additional effort the authors put into their work. However, after carefully reviewing the responses, I am (still) not convinced that the claims and, thereby, publication in Nature Communications are justified.

We would like to thank the reviewer for the comments. We have tried our best to address the questions raised by reviewer. For clarity, the reviewers' comments are in blue, and the changes we have made are highlighted in yellow in the revised manuscript.

2A. - I believe that 3D printing is possible with the method, but it has not been demonstrated. Pillars of 2-3 droplets are not “3D” and the quality needs to be significantly improved to justify the claim (as opposed to the authors, I do believe that it should be within the scope of this paper).

Response: The micropillar shown in the manuscript (#0.012, Figure 5h, main text) is having a height of 2 mm. For achieving this height, approximately 32 to 35 drops are required when using an ink suspended with ~ 44% ZrO₂ nanoparticles. Due to the ability of drop impact printing (DIP) technique to print high mass loading inks without clogging, fewer number of drops are required to achieve a given thickness. This is the merit of DIP technique. The printing of 3D micro pillar is also shown in a movie in Video S8 (supplementary video file).

2B.- The comparison to existing processes is still incomprehensive.

Response: We have tried to do a detailed comparison of our technique with the existing techniques in the following table:

Table R1: Comparison of different techniques with drop impact technique.

	Drop on demand (DoD) (Piezo based) Printing ^{1,2}	Continuous (Piezo based) Printing	Electrohydrodynamic Printing ³	Acoustophoretic Printing ⁴	Drop Impact Printing, (DIP)
Resolution, μm	25	20 ⁵	10	37 ⁶	42
Ink Viscosity, mPas	3 - 35	2- 10 ⁷	~1000	0.5 - 25,000	<33
Ink Surface tension, mN/m	44 - 54	20 – 35 ⁷	NA	NA - 624	32-72
Position accuracy, μm	66 ⁵	Low	10	60	10
Mass loading (%)	<20	~10	~30	NA	71
Max particle size, μm	<0.1	<1	<1.5 ⁸	10	20
Nozzle diameter, μm	5	60	5-1000	2000	25 – 533
Droplet detachment mechanism	Pressure waves	Pressure waves	Electrohydrodynamic instability	Acoustic focusing	Cavity collapse induced pressure wave
Energy source	Piezo/thermal driven	Piezo/thermal driven	Voltage driven (Voltage<10kV)	Acoustic radiation pressure	Gravity driven
Working distance, mm	1	5-20 ²	4.5 – 5.5	3.15	1-5
Ink palette	Mostly all inks	Conductive charged inks	Conductive inks, viscous inks	Mostly all inks, high viscous inks	Mostly all inks
Drop volume	1pL - 8pL	4pL -1.76nL ⁵	2pL ⁷ - 135pL ⁹	20nL – 800nL	38 pL – 463 nL
Drop on Demand	Yes	No	No	Yes	Yes
Cost (printhead), \$	100-1000	100-1000	Low cost	NA	9.4
Nozzle Clogging	Yes	Yes	Can be minimized up to some extent	Less clogging	No

NA - Data is not available

The above comparison is shown for three basic types of printing techniques namely DOD inkjet, continuous inkjet, and electrohydrodynamic inkjet. For some parameters, comparison with the acoustophoretic technique is also shown (based on available data in literature). Significant improvement in printing parameters for DIP has been shown in red letters in the above table. This comparison clearly shows the merits and novelty of the DIP technique.

2C.- The authors claim a broad range of printable materials, but the required recyclability of the inks limits the materials palette significantly (e.g., no solvent-based inks, no expensive inks, ...).

Response: Here the reviewer has raised concern that after impact, the remaining ink will continuously evaporate, and recyclability will be an issue. When we have used the term “recyclability”, we have meant that we collect and reuse the liquid in the droplet immediately. To demonstrate this capability, we have designed and implemented a setup as shown in Figure R1. A small peristaltic pump is integrated with an enclosed reservoir to supply the droplets for impact. The ink reservoir is in a partially sealed environment inside a cabinet. The mesh is installed with a slight tilt ($< 5^\circ$) to ensure that the droplet is captured in the reservoir after impact. This slight tilt does not affect the cavity formation phenomena or the droplet ejection (unpublished results). The environment in this partially sealed reservoir can be easily controlled to minimize exposure to oxygen and loss of solvent due to evaporation. Thus, the printing process can continue without the addition of makeup solvent. Further improvement of the setup to achieve a completely sealed environment is possible by following the report of Fish et.al.¹⁰. The use of such design thus ensures applicability of droplet impact printing with expensive and solvent-based inks.

Moreover, the evaporated volume from a single drop has been studied previously and found to be very small for first few minutes^{11–13}. For a drop of volume 2 μL with a vapor pressure 2.3 kPa, the evaporated volume in first 100 ms (timescale of the drop impact process) is ~ 0.0005 microliter¹². Loss in the droplet volume is negligible ($\sim 10^{-2}$ %). Thus, evaporation from the drop during printing and before recollection is minimal. Further, a detailed review of ink material palette used for different applications (Table R2, R3 and R4) shows that mostly inks are formulated to match the surface tension and viscosity range of the used technique^{7,14–16} ($1 < Z < 14$ for inkjet printers). The vapor pressure of formulated ink is usually kept between 0.3 to 5 kPa^{17–19}. This is done to reduce evaporation. Thus, drop impact printing technique ($3 < Z < 300$) can handle inks commonly used in other printing techniques. Drop impact printing lacks the capability to handle very high viscosity inks. However, other capabilities of the drop impact printing provide it a wider material palette than several other technologies.

Figure R1: Solidworks design file and lab scale prototype for Drop impact printing recycle unit. The unit consists of an ink reservoir connected with a peristaltic pump, sieve and a cabinet sealing the ink reservoir to avoid any loss of ink due to evaporation.

Table R2: Literatures related to additive manufacturing for different techniques.

Author ^{Citation}	Technique	Surface Tension, mN/m	Maximum Viscosity, mPas	Ink
Blazdell et. al. ²⁰	Continuous Inkjet; (BIO.DOT)	-NA-	< 0.01	Ethanol + Dispersant + Polyvinyl butyral, for charging ink (ammonia and acetic acid)
Xiang et. al. ²¹	Drop on Demand Piezo Inkjet	21.2	0.00951	Propanol (86.8%) + Ethanol (9.6%) + dispersant
Teng and Edirisinghe ²²	Continuous Inkjet; (BIO.DOT)	~25	0.00164	Ammonium nitrate + Dispersant + Ethanol
Slade and Evans ²³	Thermal Inkjet printer	52	0.003	Water (83.3%) + Polyethyleneglycol (6.7%)
Song et. al. ²⁴	Continuous Inkjet; (BIO.DOT)	>25	0.0054	Methylated spirit, Poly(vinyl butyral) binder, Dispersant, Plasticizer
Mott et. al. ²⁵	Piezoelectric IBM inkjet	~22	0.0062	Propanol (85.5%) + Ethanol (9.5%)
Windle ²⁶	Epson Stylus 500C	~72	0.0036	Polyvinyl alcohol (1.2%) + Water (96.3%)
Seerden et. al. ²⁷	Piezo inkjet printing	~25	0.0038	Paraffin wax (%) + Dispersant (%) + Organic compounds
Zhao et. al. ²⁸	Piezo inkjet printing	-NA-	-NA-	Octante (56.89%) + Isopropyl alcohol (14.21%) + Wax (2.84%) + Dispersant (11.85%)
Kosmala et. al. ²⁹	Drop on Demand Piezoelectric inkjet	-NA-	<0.0033	Water (52.75%) + Pluronic F127 (2.25%)
Lee et. al. ¹⁴	Piezoelectric inkjet	-NA-	0.0017	Water (96%) + Tego Dispersant (2%)

Zhou et. al. ³⁰	Piezoelectric Epson inkjet	~23.8	0.0013	Cyclohexane and dodecane (1:1)
Nallan et. al. ¹⁵	Custom Drop on Demand Piezoelectric inkjet	-NA-	~ 0.023	Hexane (8%) and α -terpineol (72%)
Salari et. al. ¹⁶	Piezoelectric inkjet (HP 61 cartridge)	26	0.0104	Isopropyl alcohol (40) + α -Terpineol (45)
Lee et. al. ³¹	Electrohydrodynamic Printing	NA	NA	Toulene + Dispersant
Park et. al. ³²	Electrohydrodynamic Printing	NA	~0.002	Polyethyleneglycol methyl ether (7.5%) + Water (67.5%)
Lee et. al. ³³	Electrohydrodynamic Printing	48	~0.0173	Ethylene glycol (80%)
Yu et. al. ³⁴	Electrohydrodynamic Printing	32	0.005	Toulene + Dispersant

Table R3: Literatures related to bio-based printing applications for different techniques.

Author ^{Citation}	Technique	Surface Tension, mN/m	Maximum Viscosity, mPas	Solvent (%)
A. Negro et. al. ³⁵	Inkjet printer	~53	0.005	Alginate (0.5% w/v) Polyethylene glycol (3%w/v) + enzymes + Ethylenediaminetetraacetic acid (EDTA, 0.66 mM)
E. Cheng et. al. ³⁶	Inkjet printer	~53	0.005	Phosphate buffered saline, BSA, MCF-7 breast cancer cells (1,500,000 cells/mL)
Tao Xu et al. ³⁷	Thermal Drop on Demand Bioprinter	-NA-	-NA-	Sodium alginate, gluronic acid, phosphate buffered saline, Beta-TC6 cells
L. Gasperini et. al. ³⁸	Electrohydrodynamic Bioprinter	-NA-	-NA-	Medium, Phosphate-buffered saline (PBS), Trypsin/ Ethylenediaminetetraacetic acid (EDTA) and the alginate solution

Table R4: Literatures related to food and pharmaceutical applications for different techniques.

Author ^{Citation}	Technique	Surface Tension, N/m	Maximum Viscosity, Pas	Solvent (%)
Sandler et. al. ³⁹	Inkjet printing	52	0.0031	Paracetamol, caffeine, and theophylline in propylene glycol -water solution (30:70, v/v%)
Lee et. al. ⁴⁰	Piezoelectric Inkjet	35.4	0.00599	Poly(lactic-co-glycolic acid) (100mg/mL), paclitaxel (PTX) (10mg/mL), dimethylacetamide (DMAc)
Gu et. al. ⁴¹	Piezoelectric Inkjet	43.5	0.0077	6 wt.% Poly(lactic-co-glycolic acid), 2 wt.% rifampicin (RFP) and 2 wt.% biphasic calcium phosphate (BCP)
Cate et. al. ⁴²	Inkjet printing	-NA-	-NA-	Printing ink: sodium alginate (3w%), and calcium Chloride (5 w%) in dematerialized water Encapsulating material: linseed oil, carrageenan (3w%) dissolved in dematerialized

2D.- There are too many claims that various things could be possible after further optimization of the technique (including the aforementioned 3D printing or the printing of higher-viscosity inks). I agree that some of these things seem feasible, but others are not so obvious to me.

Response: The claims of this work are as follows

1. High mass loading (71%) printing capability
2. Printing suspensions solution of higher particle sizes ($D_{nozzle}/D_{particle} \sim 4$)
3. Satellite free droplet printing without any specific optimization^{43,44}.
4. Large droplet printing range (38 pL – 463 nL) in a single setup.
5. High accuracy printing as high as 10 μm (longitudinal direction).
6. Cavity-collapse driven hydrodynamic singularity is recognized as the reason for the ejection of droplets. This mechanism ensures satellite free droplet generation.
7. For certain sieve dimensions we have identified interface recoil from the pores as a completely new mode of cavity formation, which is unique to impact on sieves.

Major claims in terms of applications

1. Single cell printing (Printing of satellite free viscoelastic biological suspended solution)
2. Large arrays of biological suspended solution printing for research purposes. (printing larger cells is uniqueness of this technique)
3. Printing of electronic materials on various substrates (including flexible)
4. Additive manufacturing of 3D structures (Although the throughput is less as compared to other techniques, but parallelization of multiple nozzle and high mass loading printing ability will ensure to fabricate structures at equivalent timescale).

We believe we have made justified claims based on the experimental data. We agree to the fact that drop impact printing does not have the widest possible material palette. Unlike acoustophoretic printing & laser based printing, where printing with high viscosity inks⁴ has been demonstrated, drop impact printing is unable to work with very high viscosity liquids. However, drop impact printing technique has many advantages over the existing printing techniques as explained above. We agree with the reviewer that printing of high viscosity ink is not trivial and further improvements (augmentation) in the setup is required to achieve this. We have modified the text in the manuscript to address this aspect.

E.- The authors claim that clogging, e.g, in Inkjet Printing (again, the only comparison is Inkjet Printing) occurs due to the evaporation of solvents. However, if solvent-based inks are used here, the inks won't be recyclable.

Response: Evaporation from solvent or water-based inks is an inevitable phenomenon. It affects all printing process. But we should carefully consider the evaporated volume within the relevant timescale and its effect on the process. For nozzle-based printing, meniscus is continuously maintained at the nozzle tip. Hence, the time available for the evaporation process is longer. For a nozzle tip with 100 μm diameter, a volume loss of ~ 3 pL is equivalent to a meniscus displacement of 100 μm . Thus, the ink evaporation at nozzle tip will results in significant increase in particle concentration and subsequent accumulation. In contrast, for drop impact printing, there is no continuous contact of ink with nozzle (i.e mesh pore). The contact time of ink with sieve is approximately 10 ms. The overall process, from release to recollection in reservoir is completed in less than a second. As described above, the loss in drop volume in this timescale is negligible ($\sim 10^{-2}$ %). Finally, we would like to highlight that clogging is not only due to evaporation. It is also often due to contaminants in the ink. A single contaminant with size in the order of nozzle tip, will significantly increase to possibility of its clogging. The short contact time of the drop impact process ensures that the sieve pore is immune to such contaminants. This is a unique advantage of drop impact printing technique as compared to the existing techniques.

Further a detailed review of nozzle clogging phenomena for different techniques is summarized below. We have set the maximum mass loading as a deciding entity up to which the corresponding technique can print suspension solution without clogging. The technique that can print higher mass loading is less prone to nozzle clogging.

Table R5: Literatures citing mass loading and other parameters of ink used by different techniques.

Journal Author ^{Citation}	Technique	Nanoparticle Type	Mass loading (%)	$D_{\text{nozzle}}/D_{\text{particle}}$
Blazdell et. al. ²⁰	Continuous Inkjet; (BIO.DOT)	Zirconia 5 wt % Yttrium oxide	5.3 (vol%)	-NA-
Xiang et. al. ²¹	Drop on Demand Piezo Inkjet	Titanium dioxide	5.3 (vol%)	260
Teng and Edirisinghe ²²	Continuous Inkjet; (BIO.DOT)	Zirconia	2.4 (vol%)	650
Slade and Evans ²³	Thermal Inkjet printer	Zirconia	10 (vol%)	250
Song et. al. ²⁴	Continuous Inkjet; (BIO.DOT)	Zirconia 5 wt % Yttrium oxide	5 (vol%)	600

Mott et. al. ²⁵	Piezoelectric IBM inkjet	Zirconia, Carbon	2.5 (vol%)	325
Windle ²⁶	Epson Stylus 500C	Lead zirconate titanate (PZT)	2.2 (vol%)	375
Seerden et. al. ²⁷	Piezo inkjet printing	Aluminium oxide + Zirconia	30 (wt%)	187.5
Zhao et. al. ²⁸	Piezo inkjet printing	Zirconia	14.21 (vol%)	111.11
Kosmala et. al. ²⁹	Drop on Demand Piezo Inkjet	Silver ink	45 (wt%)	2100
Lee et. al. ¹⁴	Piezoelectric inkjet	Zinc oxide	2 (vol%)	252.94
Zhou et. al. ³⁰	Piezoelectric Epson inkjet	Silver ink	20 (wt%)	5000
Nallan et. al. ¹⁵	Custom Drop on demand Piezoelectric inkjet	Gold	20 (wt%)	24000
Salari et. al. ¹⁶	Piezoelectric inkjet (HP 61 cartridge)	Zirconia 3 mol % Yttrium oxide	15 (wt%)	33.33
Lee et. al. ³¹	Electrohydrodynamic Printing	Silver	30 (wt%)	20000
Park et. al. ³²	Electrohydrodynamic Printing	PEDOT:PSS	25 (wt%)	1000
Lee et. al. ³³	Electrohydrodynamic Printing	Silver	20 (wt%)	9000
Yu et. al. ³⁴	Electrohydrodynamic Printing	Silver	30 (wt%)	60000
Wu et. al. ⁴⁵	Electrohydrodynamic Printing	Lead loaded stannic oxide	4.7 (wt%)	200

From Table R5 we can conclude that the general mass loading (%) for printing is around 20 % and the highest achieved is 45% (after modification of printhead). The ratio of nozzle to particle size is above 100 for all the existing techniques. In comparison, drop impact printing technique can print as high as 71% mass loading and ratio of $D_{\text{nozzle}}/D_{\text{particle}}$ is ~ 4 (printing of larger particle size).

2F.- About 6% of the mass loading gets lost in the process (and likely settles on the sieve?), which I find significant and, depending on the application, intolerable (as opposed to the authors' opinion).

Response: We agree with the reviewer that there is a discrepancy of 5% in terms of mass loading. This loss in mass loading may be due to settling of nanoparticles inside the reservoir (syringe) or pipes. However, the important point is that this reduction of mass loading by 5% is consistent throughout the experiment. We printed 50 droplets with 71% mass loading. After drying the mass loading evaluated from the remaining solid mass was $66\% \pm 1.5\%$. This shows that although there is a loss in mass loading in drop impact printing technique, the same is consistent throughout the experiments. As the reduction is consistent, this reduction can be

calibrated in practical applications and printing can be performed without worrying about drop to drop variation. We would like to mention here that even the deposited mass loading of 66% is significantly higher than previous demonstrations.

2G.- The mother-drop formation at a height above the sieve, driven by gravitational forces, seems finicky and rather limiting in terms of the throughput speed (the authors mention 6 drops per second). For an actual 3D printing process, more elegant solutions are required. These solutions might be more costly, diminishing their claim to be significantly cheaper than other methods.

Response: Authors believe that gravitational force driven drop impact on sieve makes this technique unique compared to others. Similar to the existing techniques in practice, drop impact printing also requires optimisation to achieve single drop printing zone. For piezo based techniques, there is always a need to optimize the time period of the waveform and voltage required to eject a single drop⁴⁶⁻⁴⁸. This varies for different nozzle and inks. Same is for EHD printers⁴⁹ and continuous inkjet printers⁵⁰. Using gravity driven impact mode for printing, we are making the setup simpler, easy to handle and it requires less optimization parameters (only height manipulation is required here).

We completely agree that drop impact technique has low throughput in terms of droplets per second and cannot be beneficial where high throughput is required. But at the same time this technique makes up in terms of printing thicker lines and supporting multiple impact locations. This is a very important aspect because unlike in other techniques where multiple pass not only decrease the resolution but also the printing quality⁵¹. This technique ensures thick layer by layer printing in one go due to its capability to handle high mass loading.

We agree with the reviewer that if a more complicated setup is required for specific applications the setup cost will increase. But still the cost will be low as compared to other techniques due the use of single 3 x 2 cm² sieve as a multiple nozzle system. (one 3*2 cm² mesh patch cost around \$9.4). Use of the sieve completely eliminates all kinds of nozzle design and fabrication complications⁵²⁻⁵⁵. Also during use, nozzle cleaning^{56,57} will not be necessary frequently in drop impact printing technique. We believe all these will significantly reduce the cost of use.

Reviewer #3 (Remarks to the Author):

3A. The comments are addressed. The only remaining question is on recycling of the liquid as 99% of the liquid should be recycled on each impact. This is a challenge in real implementation of this method.

Response: We agree with the reviewer that in this technique we need to reuse the inks after impact. When we have used the term “recyclability”, we have meant that we collect and reuse the excess liquid in the droplet immediately. Authors believe that this can be implemented effectively. A lab scale setup along with design file of recycle unit is shown in Supplementary Figure S18. A video of the operational printing setup is shown in Video S10. In this setup, a small peristaltic pump is integrated with an enclosed reservoir to supply the droplets for impact. The ink reservoir is in a partially sealed environment inside a cabinet. The mesh is installed with a slight tilt ($< 5^\circ$) to ensure that the droplet is captured in the reservoir after impact. This slight tilt does not affect the cavity formation phenomena or the droplet ejection (unpublished results). The environment in this partially sealed reservoir can be easily controlled to minimize exposure to oxygen and loss of solvent due to evaporation. The compact setup shown here reduces evaporation of the ink and thus the ink can be continually used without requiring additional solvent. This is similar to the report of Fish et. al. ¹⁰, where a closed sealed printing setup has been used to minimize loss of ink due to evaporation⁵⁸.

Drop impact printing setup with recycle unit. The labelling are as follows:

1. Ink reservoir
2. Peristaltic pump (pumping ink to syringe)
3. Mother droplet generation using a syringe
4. Automated z stage for height manipulation
5. Holding platform for mesh
6. Substrate holder fixed with xyz automatic stage
7. High-speed camera
8. Diffused light
9. Sealed cabinet

References

1. Derby, B. Inkjet printing of functional and structural materials: fluid property requirements, feature stability, and resolution. *Annual Review of Materials Research* **40**, 395–414 (2010).
2. Castrejon-Pita, J. R. *et al.* Future, opportunities and challenges of inkjet technologies. *Atomization and sprays* **23**, (2013).
3. Ball, A. K., Das, R., Roy, S. S., Kisku, D. R. & Murmu, N. C. Experimentation modelling and optimization of electrohydrodynamic inkjet microfabrication approach: a Taguchi regression analysis. *Sādhanā* **44**, 167 (2019).
4. Foresti, D. *et al.* Acoustophoretic printing. *Science Advances* **4**, eaat1659 (2018).
5. Park, J. A. *et al.* Freeform micropatterning of living cells into cell culture medium using direct inkjet printing. *Scientific Reports* **7**, 14610 (2017).
6. Demirci, U. & Montesano, G. Single cell epitaxy by acoustic picolitre droplets. *Lab on a Chip* **7**, 1139–1145 (2007).
7. Magdassi, S. *The chemistry of inkjet inks*. (World Scientific, 2010).
8. Huang, Y. *et al.* Study effects of particle size in metal nanoink for electrohydrodynamic inkjet printing through analysis of droplet impact behaviors. *Journal of Manufacturing Processes* (2020).
9. Guo, L., Duan, Y., Huang, Y. & Yin, Z. Experimental Study of the Influence of Ink Properties and Process Parameters on Ejection Volume in Electrohydrodynamic Jet Printing. *Micromachines* **9**, 522 (2018).
10. Fish, L. A. Sealed printing mechanism using highly volatile inks. (1971).
11. Cantú, A. A. A study of the evaporation of a solvent from a solution—application to writing ink aging. *Forensic science international* **219**, 119–128 (2012).
12. He, P. & Derby, B. Controlling Coffee Ring Formation during Drying of Inkjet Printed 2D Inks. *Advanced Materials Interfaces* **4**, 1700944 (2017).
13. Feng, J. Q. Vapor transport of a volatile solvent for a multicomponent aerosol droplet. *Aerosol Science and Technology* **49**, 757–766 (2015).
14. Lee, A., Sudau, K., Ahn, K. H., Lee, S. J. & Willenbacher, N. Optimization of experimental parameters to suppress nozzle clogging in inkjet printing. *Industrial & engineering chemistry research* **51**, 13195–13204 (2012).
15. Nallan, H. C., Sadie, J. A., Kitsomboonloha, R., Volkman, S. K. & Subramanian, V. Systematic design of jettable nanoparticle-based inkjet inks: Rheology, acoustics, and jettability. *Langmuir* **30**, 13470–13477 (2014).
16. Salari, F., Badihi Najafabadi, A., Ghatee, M. & Golmohammad, M. Hybrid additive manufacturing of the modified electrolyte-electrode surface of planar solid oxide fuel cells. *International Journal of Applied Ceramic Technology* (2020).
17. Acitelli, M. A., Merz III, C. J., Rose Jr, F. M. & Smith, J. C. Nonaqueous thermaljet ink compositions. (1991).
18. Zou, W. K., Wang, X., Woodcock, C., Dong, Q. Q. & Xiao, F. Ink jet ink composition. (2002).
19. Özkol, E., Ebert, J., Uibel, K., Wätjen, A. M. & Telle, R. Development of high solid content aqueous 3Y-TZP suspensions for direct inkjet printing using a thermal inkjet printer. *Journal of the European Ceramic Society* **29**, 403–409 (2009).
20. Blazdell, P. F., Evans, J. R. G., Edirisinghe, M. J., Shaw, P. & Binstead, M. J. The computer aided manufacture of ceramics using multilayer jet printing. *Journal of materials science letters* **14**, 1562–1565 (1995).

21. Xiang, Q. F., Evans, J. R. G., Edirisinghe, M. J. & Blazdell, P. F. Solid freeforming of ceramics using a drop-on-demand jet printer. *Proceedings of the Institution of Mechanical Engineers, Part B: Journal of Engineering Manufacture* **211**, 211–214 (1997).
22. Teng, W. D. & Edirisinghe, M. J. Development of ceramic inks for direct continuous jet printing. *Journal of the American Ceramic Society* **81**, 1033–1036 (1998).
23. Slade, C. E. Freeforming ceramics using a thermal jet printer. *Journal of Materials Science Letters* **17**, 1669–1671 (1998).
24. Song, J. H., Edirisinghe, M. J. & Evans, J. R. G. Formulation and multilayer jet printing of ceramic inks. *Journal of the American Ceramic Society* **82**, 3374–3380 (1999).
25. Mott, M., Song, J. & Evans, J. R. G. Microengineering of ceramics by direct ink-jet printing. *Journal of the American Ceramic Society* **82**, 1653–1658 (1999).
26. Windle, J. & Derby, B. Ink jet printing of PZT aqueous ceramic suspensions. *Journal of materials science letters* **18**, 87–90 (1999).
27. Seerden, K. A. M. *et al.* Ink-jet printing of wax-based alumina suspensions. *Journal of the American Ceramic Society* **84**, 2514–2520 (2001).
28. Zhao, X., Evans, J. R. G., Edirisinghe, M. J. & Song, J. H. Ink-jet printing of ceramic pillar arrays. *Journal of materials science* **37**, 1987–1992 (2002).
29. Kosmala, A., Zhang, Q., Wright, R. & Kirby, P. Development of high concentrated aqueous silver nanofluid and inkjet printing on ceramic substrates. *Materials Chemistry and Physics* **132**, 788–795 (2012).
30. Zhou, X., Li, W., Wu, M., Tang, S. & Liu, D. Enhanced dispersibility and dispersion stability of dodecylamine-protected silver nanoparticles by dodecanethiol for ink-jet conductive inks. *Applied surface science* **292**, 537–543 (2014).
31. Lee, D. Y., Hwang, E. S., Yu, T. U., Kim, Y.-J. & Hwang, J. Structuring of micro line conductor using electro-hydrodynamic printing of a silver nanoparticle suspension. *Applied Physics A* **82**, 671–674 (2006).
32. Park, J.-U. *et al.* High-resolution electrohydrodynamic jet printing. *Nature Materials* **6**, 782–789 (2007).
33. Lee, D.-Y., Shin, Y.-S., Park, S.-E., Yu, T.-U. & Hwang, J. Electrohydrodynamic printing of silver nanoparticles by using a focused nanocolloid jet. *Applied Physics Letters* **90**, 81905 (2007).
34. Yu, J. H., Kim, S. Y. & Hwang, J. Effect of viscosity of silver nanoparticle suspension on conductive line patterned by electrohydrodynamic jet printing. *Applied physics A* **89**, 157–159 (2007).
35. Negro, A., Cherbuin, T. & Lutolf, M. P. 3D Inkjet Printing of Complex, Cell-Laden Hydrogel Structures. *Scientific Reports* **8**, 17099 (2018).
36. Cheng, E., Ahmadi, A. & Cheung, K. C. Investigation of the hydrodynamics of suspended cells for reliable inkjet cell printing. in *ASME 2014 12th International Conference on Nanochannels, Microchannels, and Minichannels collocated with the ASME 2014 4th Joint US-European Fluids Engineering Division Summer Meeting* (American Society of Mechanical Engineers Digital Collection, 2014).
37. Xu, T., Kincaid, H., Atala, A. & Yoo, J. J. High-throughput production of single-cell microparticles using an inkjet printing technology. *Journal of Manufacturing Science and Engineering* **130**, (2008).
38. Gasperini, L., Maniglio, D., Motta, A. & Migliaresi, C. An electrohydrodynamic bioprinter for alginate hydrogels containing living cells. *Tissue engineering part C: Methods* **21**, 123–132 (2015).
39. Sandler, N. *et al.* Inkjet printing of drug substances and use of porous substrates-towards

- individualized dosing. *Journal of pharmaceutical sciences* **100**, 3386–3395 (2011).
40. Lee, B. K. *et al.* Fabrication of drug-loaded polymer microparticles with arbitrary geometries using a piezoelectric inkjet printing system. *International journal of pharmaceuticals* **427**, 305–310 (2012).
 41. Gu, Y. *et al.* Inkjet printed antibiotic-and calcium-eluting bioresorbable nanocomposite micropatterns for orthopedic implants. *Acta biomaterialia* **8**, 424–431 (2012).
 42. Pieterse, G. *et al.* Novel encapsulation technology for the preparation of core-shell microparticles. *Journal of controlled release: official journal of the Controlled Release Society* **148**, e8-9 (2010).
 43. He, B., Yang, S., Qin, Z., Wen, B. & Zhang, C. The roles of wettability and surface tension in droplet formation during inkjet printing. *Scientific reports* **7**, 1–7 (2017).
 44. Yang, Q. *et al.* Rayleigh instability-assisted satellite droplets elimination in inkjet printing. *ACS applied materials & interfaces* **9**, 41521–41528 (2017).
 45. Wu, H., Yu, J., Cao, R., Yang, Y. & Tang, Z. Electrohydrodynamic inkjet printing of Pd loaded SnO₂ nanofibers on a CMOS micro hotplate for low power H₂ detection. *AIP Advances* **8**, 55307 (2018).
 46. Wijshoff, H. The dynamics of the piezo inkjet printhead operation. *Physics reports* **491**, 77–177 (2010).
 47. Kim, Y. *et al.* The Effects of Driving Waveform of Piezoelectric Industrial Inkjet Head for Fine Patterns. in *2006 1st IEEE International Conference on Nano/Micro Engineered and Molecular Systems* 826–831 (2006). doi:10.1109/NEMS.2006.334905
 48. Hoath, S. D. *Fundamentals of inkjet printing: the science of inkjet and droplets.* (John Wiley & Sons, 2016).
 49. Lee, A., Jin, H., Dang, H.-W., Choi, K.-H. & Ahn, K. H. Optimization of experimental parameters to determine the jetting regimes in electrohydrodynamic printing. *Langmuir* **29**, 13630–13639 (2013).
 50. Desai, S. & Lovell, M. Statistical Optimization of Process Variables In A Continuous Inkjet Process—A Case Study. *International Journal of Industrial Engineering: Theory, Applications and Practice* **15**, 104–112 (2008).
 51. Mavuri, A., Mayes, A. G. & Alexander, M. S. Inkjet Printing of Polyacrylic Acid-Coated Silver Nanoparticle Ink onto Paper with Sub-100 Micron Pixel Size. *Materials* **12**, 2277 (2019).
 52. Lim, J.-H. *et al.* Investigation of reliability problems in thermal inkjet printhead. in *2004 IEEE International Reliability Physics Symposium. Proceedings* 251–254 (2004). doi:10.1109/RELPHY.2004.1315332
 53. Lee, K. I. *et al.* Multi nozzle electrohydrodynamic inkjet printing head by batch fabrication. in *2013 IEEE 26th International Conference on Micro Electro Mechanical Systems (MEMS)* 1165–1168 (2013). doi:10.1109/MEMSYS.2013.6474458
 54. Kim, Y., Son, S., Choi, J., Byun, D. & Lee, S. Design and fabrication of electrostatic inkjet head using silicon micromachining technology. *Journal of Semiconductor Technology and Science* **8**, 121–127 (2008).
 55. Le, H. P. Progress and trends in ink-jet printing technology. *Journal of Imaging Science and Technology* **42**, 49–62 (1998).
 56. Anderson, D. G., Claflin, A. J. & Chinnici, J. Ultrasonic liquid wiper for ink jet printhead maintenance. (1996).
 57. Isayama, T. & Kubo, K. Automatic nozzle cleaning system for ink ejection printer. (1977).
 58. Gutmann, O. *et al.* Fast and reliable protein microarray production by a new drop-in-drop technique. *Lab on a Chip* **5**, 675–681 (2005).

We have made attempts to address the points raised by the reviewers. Once again, we wish to express our gratitude to the reviewers for their careful reading of our manuscript, and for their comments and helpful suggestions. We would like to thank the editor for giving us time to address all the points.

We hope the revised manuscript will be acceptable for publication in your highly esteemed journal.

Yours faithfully,

Dr. Prosenjit Sen
Associate Professor
Centre for Nanoscience and Engineering
Indian Institute of Science
Bangalore, India 560012

REVIEWER COMMENTS

Reviewer #2 (Remarks to the Author):

I thank the authors for addressing my comments and concerns, and believe the quality of the manuscript is sufficient for publication in Nature Communications after addressing the following, minor comments.

- I appreciate the efforts of the authors building an "ink recycling system". Unfortunately, I am still not convinced that the inks are (fully) recoverable. The amount of evaporation is, as stated, small, but given. Achieving a completely sealed system is possible, but diminishes one of their other key advantages – the cost and simplicity of the system. Further, I strongly suspect that evaporation might not be the only factor of importance, as there might be changes in the effective mass loading due to the impact, etc. One way to prove this point would be to eject and recycle the totality of the ink in the reservoir, repeat it several times, and compare the mass loading, rheological properties, etc. to the original batch. However, I am fine with moving on, if the authors remove or significantly reduce the statements on the general recyclability of the inks.

- I recommend adding tables R1-R5 to the supporting information, too.

- Table R1:

-- The table seems to favor the presented method, as it lacks some parameters critical to 3D printing, such as throughput speed.

-- The effective mass loading is 66%, due to the loss of 5% in the process.

-- Re. positional accuracy: In the manuscript, the authors write " $\sim 30 \mu\text{m}$ in the lateral direction and $\sim 10 \mu\text{m}$ in the longitudinal direction" – shouldn't it be the "worst case" scenario?

-- To my understanding, in acoustophoretic printing, the nozzle diameter can be anything that can be physically fabricated, and especially smaller than $2000 \mu\text{m}$. How did the authors come up with that number?

-- The same applies to the working distance. The nozzle can simply be placed at any height above the substrate, and not at "exactly" 3.15 mm .

Reviewer #3 (Remarks to the Author):

The comments are addressed.

Manuscript ID: NCOMMS-19-33513

“Drop Impact Printing”

Reviewer #1 (Remarks to the Author):

The authors have addressed the issue I had raised in my first review. My suggestion is to publish the paper.

We would like to thank the reviewer for providing valuable comments during the review process. This indeed helped improve the quality of the manuscript.

Reviewer #2 (Remarks to the Author):

I thank the authors for addressing my comments and concerns and believe the quality of the manuscript is sufficient for publication in Nature Communications after addressing the following, minor comments.

We would like to thank the reviewer for the positive comments and recommending our work for publication with minor changes. We have tried our best to address the points mentioned by the reviewer.

- I appreciate the efforts of the authors building an “ink recycling system”. Unfortunately, I am still not convinced that the inks are (fully) recoverable. The amount of evaporation is, as stated, small, but given. Achieving a completely sealed system is possible, but diminishes one of their other key advantages – the cost and simplicity of the system. Further, I strongly suspect that evaporation might not be the only factor of importance, as there might be changes in the effective mass loading due to the impact, etc. One way to prove this point would be to eject and recycle the totality of the ink in the reservoir, repeat it several times, and compare the mass loading, rheological properties, etc. to the original batch. However, I am fine with moving on, if the authors remove or significantly reduce the statements on the general recyclability of the inks.

Response: We agree with the reviewer that the entire ink will not be completely recoverable. However, as the evaporation rate is slow in a sealed setup, its effect on ink concentration will be minimal. Previous studies on sealed setup have shown a significant decrease in the evaporation rate of liquid^{1,2}. As mentioned by the reviewer, a sealed setup for eliminating evaporation will bring in complexity in terms of precise environment control and hence increase cost. However, such a setup will not be a general requirement. Such a system will be necessary for expensive inks with high vapor pressure. We believe that even though the current system does not eliminate evaporation completely, it will be sufficient for most applications.

We agree with the reviewer that there are other issues for printing droplets with high mass loading such as loss due to sedimentation in the reservoir. However, these effect the printing process over a long duration. Further, we are investigating stirring and vibration for reducing this effect.

As recommended by the reviewer, we have modified the statement on general recyclability in the manuscript which read as follows:

“Although the entire ink will not be recoverable due to evaporation, according to previous studies the rate of evaporation in sealed setup is very low^{48,49} (volume loss percent is 0.01% for a timescale of 100 ms) and its effect on ink concentration (% mass loading) will be significant after continuously printing for a very long duration. A sealed setup for eliminating evaporation will bring in complexity in terms of precise environment control and hence increase cost. However, such a setup will not be a general requirement. Such a system will be necessary for expensive inks with high vapor pressure. We believe that even though the current system does not eliminate evaporation completely, it will be sufficient for most applications.”

- I recommend adding tables R1-R5 to the supporting information, too.

Response: We have added the tables in supporting information file and referred the same in the main text.

- Table R1:

-- The table seems to favor the presented method, as it lacks some parameters critical to 3D printing, such as throughput speed.

Response: The throughput (drops/sec) of drop impact printing technique has already been mentioned in the main text. However, an important aspect of our technique is also printing thicker drops or lines. Therefore, a new row has been added in Supplementary Table S7 showing the ratio of drop height/drop width for a single printed drop. This shows the capability of drop impact printing technique to print thicker lines as compared to other existing techniques.

-- The effective mass loading is 66%, due to the loss of 5% in the process.

Response: In the entire manuscript the mass loading percentage that we have mentioned is the mass loading of the ink in the reservoir (syringe). For clarity we have further added a statement in the manuscript about the effective mass loading of the deposited droplet after drying which read as flows:

“The average mass loading of the remaining solid mass for a batch of 50 printed drops after drying was found to be $66\% \pm 1.5\%$. This discrepancy of 5% in terms of mass loading may be due to the settling of nanoparticles inside the reservoir (syringe) or pipe during the printing process.”

-- Re. positional accuracy: In the manuscript, the authors write “~ 30 μm in the lateral direction and ~ 10 μm in the longitudinal direction” – shouldn't it be the “worst case” scenario?

Response: Yes, the reported data in the manuscript is the worst-case scenario and the same has been mentioned in the main text.

-- To my understanding, in acoustophoretic printing, the nozzle diameter can be anything that can be physically fabricated, and especially smaller than 2000 μm . How did the authors come up with that number? The same applies to the working distance. The nozzle can simply be placed at any height above the substrate, and not at “exactly” 3.15 mm.

Response: We would like to thank the reviewer for this point. As recommended by the reviewer, we have corrected the same in the manuscript.

Reviewer #3 (Remarks to the Author):

The comments are addressed.

We would like to thank the reviewer for providing valuable comments during the review process. This indeed helped improve the quality of the manuscript.

References

1. He, P. & Derby, B. Controlling coffee ring formation during drying of inkjet printed 2D inks. *Adv. Mater. Interfaces* **4**, 1700944 (2017).
2. Fish, L. A. Sealed printing mechanism using highly volatile inks. (1971).

We have made attempts to address the points raised by the reviewer. Once again, we wish to express our gratitude to the reviewers for their careful reading of our manuscript, and for their

comments and helpful suggestions. We would like to thank the editor for giving us time to address all the points.

We hope the revised manuscript will be acceptable for publication in your highly esteemed journal.

Yours faithfully,

Dr. Prosenjit Sen

Associate Professor

Centre for Nanoscience and Engineering

Indian Institute of Science

Bangalore, India 560012